# On the inadequacy of optimizing alignment and uniformity in contrastive learning of sentence representations

**Zhijie Nie**[1,4]**, Richong Zhang**[1,2]***, Yongyi Mao**[3]
[1]SKLSDE, Beihang University, Beijing, China
[2]Zhongguancun Laboratory, Beijing, China
[3]School of Electrical Engineering and Computer Science, University of Ottawa, Ottawa, Canada
[4]Shen Yuan Honors College, Beihang University, Beijing, China
{niezj,zhangrc}@act.buaa.edu.cn, ymao@uottawa.ca

## Abstract

Contrastive learning is widely used in areas such as visual representation learning (VRL) and sentence representation learning (SRL). Considering the differences between VRL and SRL in terms of negative sample size and evaluation focus, we believe that the solid findings obtained in VRL may not be entirely carried over to SRL. In this work, we consider the suitability of the decoupled form of contrastive loss, i.e., *alignment* and *uniformity*, in SRL. We find a performance gap between sentence representations obtained by jointly optimizing *alignment* and *uniformity* on the STS task and those obtained using contrastive loss. Further, we find that the joint optimization of *alignment* and *uniformity* during training is prone to overfitting, which does not occur on the contrastive loss. Analyzing them based on the variation of the gradient norms, we find that there is a property of "gradient dissipation" in contrastive loss and believe that it is the key to preventing overfitting. We simulate similar "gradient dissipation" of contrastive loss on four optimization objectives of two forms, and achieve the same or even better performance than contrastive loss on the STS tasks, confirming our hypothesis.[1]

## 1 Introduction

Unsupervised contrastive learning (Wu et al., 2018) is originated from visual representation learning (VRL) (Chen et al., 2020; He et al., 2020; Grill et al., 2020), and has achieved impressive performances therein. Briefly, contrastive learning forces the representation of an input instance (or "anchor") to be similar to that of an augmented view of the same instance (or "postive example") and to differ from that of some different instances (or "negative examples"). One plausible justification of this approach is that minimizing the loss of contrastive learning (or "contrastive loss") is shown to be equivalent to simultaneously minimizing an "alignment loss" and a "uniformity loss", where the former dictates the representation similarity between an instance and its positive examples, and the latter forces the representations of all instances to spread uniformly on the unit sphere in the representation space (Wang & Isola, 2020). Notably this decomposition of the contrastive loss relies on the condition that the number of negative examples participating in the contrastive loss approaches infinity. For VRL, it is arguable that such a condition holds reasonably well since usually a large number (e.g., 65536 (He et al., 2020)) of negative examples are used in training.

In recent years, contrastive learning has also been adapted to sentence representation learning (SRL), by fine-tuning the representations obtained from a pretrained language model (e.g. BERT (Devlin et al., 2018)) (Yan et al., 2021; Giorgi et al., 2021; Gao et al., 2021; Zhang et al., 2022b). Such approaches have demonstrated great performances not only for downstream classification tasks, but also for semantic textual similarity (STS) tasks. It is noteworthy that significantly contrast-

---

*Corresponding author
[1]The codes and checkpoints are released for study at https://github.com/BDBC-KG-NLP/ICLR2023-Gradient-Dissipation.git

ing VRL, which are primarily evaluated using downstream classification tasks (Russakovsky et al., 2015; Krizhevsky et al., 2009), or via "extrinsic" protocols (Chiu et al., 2016; Faruqui et al., 2016), SRL particularly emphasizes the STS tasks, or "intrinsic" protocols, when evaluating the quality of learned sentence representations (Reimers & Gurevych, 2019; Li et al., 2020; Yan et al., 2021; Zhang et al., 2022c). This is because the representations obtained from the pretrained language models have already shown strong transfer capability to downstream tasks (Devlin et al., 2018; Liu et al., 2019) while their similarities are rather poorly correlated with the human-rated similarities (Reimers & Gurevych, 2019; Li et al., 2020).

Following the justification of contrastive learning in VRL(Wang & Isola, 2020), some works (Gao et al., 2021; Zhang et al., 2022b) attribute the success of contrastive learning on the STS tasks to a good balance between *alignment* and *uniformity*. Consequently, alignment and uniformity losses are adopted widely as the key metrics for evaluating the goodness of sentence representations learned from contrastive learning (Gao et al., 2021; Zhang et al., 2022b;c;a; Klein & Nabi, 2022).

Noting that contrastive learning for SRL in fact only uses a rather small number of negative examples (e.g, 63 (Gao et al., 2021; Zhang et al., 2022b) or smaller (Zhang et al., 2022c)), in this paper, we question whether the "decomposition principle", or jointly optimizing *alignment* and *uniformity*, adequately explains the performance gain in the STS tasks brought by contrastive learning.

After extensive experiments, we find that optimization using *alignment* and *uniformity* losses produces lower performance than that with contrastive loss in the STS tasks. Moreover, this performance degradation is not reflected by the *alignment* and *uniformity* metrics. Interestingly, we also observe the same phenomenon in contrastive learning with another loss function, Decoupled Contrastive Loss (DCL) (Yeh et al., 2021), in which the optimization objectives are very similar to *alignment* and *uniformity*. Our further studies also show that training with such decoupled forms of contrastive loss cause severe overfitting, which does not occur in training with the standard contrastive loss. These observations suggest that *alignment* and *uniformity* losses might not serve suitable substitutes for contrastive loss in SRL and that the success of the standard contrastive learning for SRL can not be adequately explained in terms of *alignment* and *uniformity* properties or some delicate balance between the two.

This paper focuses on uncovering other important factors, beyond *alignment* and *uniformity*, that contribute to the success story of contrastive learning in SRL and explain the performance gap between the training scheme using the contrastive loss and those using a decoupled contrastive loss (in terms of *alignment* and *uniformity* or their equivalent). Specifically, we hypothesize that the training dynamics of the standard contrastive learning for SRL plays an essential role in its effectiveness. To that end, we decompose the gradient of the contrastive loss into an *alignment component* and a *uniformity component* and compare the norms of these two components with their counter-parts in training with the decoupled contrastive losses. Interestingly, we observe a distinct "gradient dissipation" phenomenon in training with the standard contrastive loss: the gradient signal quickly drops and vanishes as soon as the negative example is *adequately further* away from the anchor than the positive example, where the adequacy appears to be reflected by a rather moderate threshold. Notably such a phenomenon does not appear in the training schemes using a decoupled contrastive loss, when the negative sample size is small. This observation led us to believe that "gradient dissipation" plays an essential role in standard contrastive learning.

To validate this hypothesis, we construct two new loss functions, both capable of inducing "gradient dissipation" in their training dynamics. We test them experimentally and observe that indeed training with both losses gives comparable or even better performance in the STS tasks than the standard contrastive loss. Interestingly, similar to the contrastive loss, training with these new loss functions also eliminates the alleviates the overfitting problem observed in training using decouple contrastive losses. This confirms that gradient dissipation serves a key role in the success of contrastive learning for SRL and suggests that properly conditioning the training dynamics is more important than optimizing *alignment* and *uniformity* when not too many negative examples are used.

Along our development, we provide insights as to why gradient dissipation is a desirable property. We also provide additional theoretical justifications on the effectiveness of the new loss functions by showing that they are in fact upper bounds of the standard contrastive loss. Due to page limit, some results, derivations and discussions are presented in Appendix.

## 2 PRELIMINARY

### 2.1 CONTRASTIVE LEARNING IN SENTENCE REPRESENTATION LEARNING

Let $\mathbb{C}$ be a set of sentences. For each $x_i \in \mathbb{C}$, we use $x_i'$ to denote an augmented view of $x_i$. With respect to anchor $x_i$, $x_i'$ is referred to a positive example, and any $x_j$ or $x_j'$ $(j \neq i)$ is referred to as a negative example. There is an encoder mapping each $x_i$ and $x_i'$ to their representations $h_i$ and $h_i'$, which are vectors in $\mathbb{R}^d$. Considering the metric invariance, we usually normalize them to obtain $\hat{h}_i$ and $\hat{h}_i'$, which are constrained on the unit hypersphere $S^{d-1}$ centered at the origin. InfoNCE Loss (Oord et al., 2018) or "contrastive loss" for anchor $x_i$ is defined by

$$\mathcal{L}_{\text{cl}} = -\log \frac{\exp\left(\hat{h}_i^T \hat{h}_i'/\tau\right)}{\exp\left(\hat{h}_i^T \hat{h}_i'/\tau\right) + \sum_{j,j \neq i}^{N} \exp\left(\hat{h}_i^T \hat{h}_j'\right)/\tau} \tag{1}$$

where $\tau$ is the temperature hyperparameter and $N$ represents the number of negative samples. The quality of sentence representations is shown to insensitive to $N$ (Gao et al., 2021), and a small $N$, such as 63 (Gao et al., 2021; Zhang et al., 2022b) or smaller (Zhang et al., 2022c) has shown to be the same sufficient as the bigger one.

### 2.2 THE DECOUPLE VERSION OF CONTRASTIVE LOSS

*Alignment* and *uniformity* (Wang & Isola, 2020) decoupled from contrastive loss are shown to be two significant properties. The losses based on the two properties are defined as

$$\mathcal{L}_{\text{a\&u}} = (1-\lambda)\mathcal{L}_{\text{align}} + \lambda\mathcal{L}_{\text{uniform}} \tag{2}$$

$$\mathcal{L}_{\text{align}}(f;\alpha) \triangleq \mathop{\mathbb{E}}_{(\hat{h}_i,\hat{h}_i') \sim p_{\text{pos}}} \left[\|\hat{h}_i - \hat{h}_i'\|_2^\alpha\right], \quad \alpha > 0 \tag{3}$$

$$\mathcal{L}_{\text{uniform}}(f;t) \triangleq \log \mathop{\mathbb{E}}_{\substack{\text{i.i.d} \\ (\hat{h}_i,\hat{h}_j') \sim p_{\text{data}}}} \left[e^{-t\|\hat{h}_i - \hat{h}_j'\|_2^2}\right], \quad t > 0 \tag{4}$$

where $\alpha$, $t$ and $\lambda$ are three hyperparameters. In the meantime, we follow another work (Yeh et al., 2021) on decoupled contrastive learning, which removes the positive sample part from the denominator of contrastive loss:

$$\mathcal{L}_{\text{dcl}} = -\log \frac{\exp\left(\hat{h}_i^T \hat{h}_i'/\tau\right)}{\sum_{j,j \neq i}^{N} \exp\left(\hat{h}_i^T h_j'\right)/\tau} = \underbrace{-\hat{h}_i^T \hat{h}_i'/\tau}_{\text{alignment}} + \underbrace{\log \left(\sum_{j,j \neq i}^{N} \exp\left(\hat{h}_i^T h_j'\right)/\tau\right)}_{\text{uniformity}} \tag{5}$$

Note that the *alignment* parts in the two works are recognized as equivalent (Yeh et al., 2021) and we proved the *uniformity* part of them have the same lower bound (Appendix D), which corresponds to the optimization objective of Minimum Energy Problem on the hypersphere (Kuijlaars & Saff, 1998; Liu et al., 2018). Therefore, the above two decoupled forms of contrastive loss are treated equally in this paper.

### 2.3 PERFORMANCE COMPARISON

We compare the performance of contrastive loss and its two decoupled forms based on SimCSE (Gao et al., 2021), and evaluate with seven datasets on semantic textual similarity (STS) tasks and seven downstream classification datasets on transfer (TR) tasks from the SentEval toolkit (Conneau & Kiela, 2018). We report the average Spearman correlation for the STS tasks and the average accuracy for the TR tasks in Table 1 . Please refer to Appendix A for experimental details.

As can be seen from the results, we can find that (1) the performance on both STS and TR tasks is better than original pretrained models after optimizing with any one of the three loss functions; (2) the main improvement over the original pretrained models is on the STS tasks, while the improvement on the TR tasks is relatively weak; (3) the decoupled forms obtain the same or better performance on TR tasks as contrastive loss, but there still have a performance gap between them on STS tasks. These observations reflect to some extent the validity of *alignment* and *uniformity* to replace contrastive loss on TR tasks, but it also implies that there are some other factors playing a key role in the improvement of STS tasks, which have been neglected in the previous studies.

| Method | BERT$_{\text{base}}$ | | BERT$_{\text{large}}$ | | RoBERTa$_{\text{base}}$ | | RoBERTa$_{\text{large}}$ | |
| --- | --- | --- | --- | --- | --- | --- | --- | --- |
| | STS.Avg | TR.Avg | STS.Avg | TR.Avg | STS.Avg | TR.Avg | STS.Avg | TR.Avg |
| Pretrained | 56.70 | 85.34 | 54.11 | 85.52 | 56.57 | 83.20 | 53.90 | 83.76 |
| $\mathcal{L}_{\text{cl}}$ | 76.04 | 86.38 | 76.83 | 86.14 | 77.31 | 85.23 | 78.52 | 85.71 |
| $\mathcal{L}_{\text{a\&u}}$ | 72.62 | 87.52 | 74.58 | 87.77 | 72.64 | 85.09 | 73.02 | 85.68 |
| $\mathcal{L}_{\text{dcl}}$ | 71.13 | 85.18 | 72.73 | 87.06 | 73.18 | 85.48 | 72.43 | 86.05 |
| $\mathcal{L}_{\text{dcl+}}$ | 75.25 | 86.95 | 77.40 | 87.85 | 76.33 | 84.52 | 75.63 | 85.57 |
| $\mathcal{L}_{\text{mpt}}$ | 77.25 | 87.56 | 77.35 | 87.71 | 76.42 | 85.10 | **78.84** | 86.51 |
| $\mathcal{L}_{\text{met}}$ | **78.38** | **87.94** | **78.38** | 87.94 | **77.38** | **85.74** | 78.71 | 86.42 |
| $\mathcal{L}_{\text{mat}}$ | 77.76 | 88.65 | 77.76 | **88.65** | 76.95 | 85.64 | 78.82 | **87.06** |

Table 1: Performance of the optimization objectives studied in this paper. All results reported are the average value obtained from three runs. STS.Avg represents the average of Spearman correlation on seven semantic textual similarity datasets. TR.Avg represents the average of accuracy on seven downstream classification datasets.

## 3 THE INADEQUACY OF OPTIMIZING ALIGNMENT AND UNIFORMITY

### 3.1 PROBLEM ANALYSIS

To gain a deeper understanding for the above performance gap, we record several metrics during the training and evaluation process, the images of which are shown in Figure 1.

Figure 1a shows the $\mathcal{L}_{\text{uniform}}$-$\mathcal{L}_{\text{align}}$ scatterplot of sentence representations on STS-B (Cer et al., 2017) development set, where the colors represent the average Spearman correlation of the STS tasks. It is important to note that the scatterplots of this style are widely used in recent works to show that the proposed method is able to achieve a better balance between *alignment* and *uniformity* than other methods (Gao et al., 2021; Zhang et al., 2022a;b;c; Klein & Nabi, 2022). Indeed, the scatterplot shows that the sentence representations whose ($\mathcal{L}_{\text{uniform}}$, $\mathcal{L}_{\text{align}}$) located in the middle of the image perform better than those whose location in the top left or bottom right. However, the performance gap between different loss functions are not reflected by the locations in this figure. Specifically, the sentence representations pretrained by $\mathcal{L}_{\text{cl}}$ and its decoupled forms can obtain the almost same values of *alignment* and *uniformity*, but an obvious performance gap on the STS tasks.

Figure 1b shows the numerical variation of $\mathcal{L}_{\text{cl}}$, $\mathcal{L}_{\text{align}}$ and $\mathcal{L}_{\text{uniform}}$ on the training set and development set during the optimization process with $\mathcal{L}_{\text{cl}}$. In terms of trends, all metrics on both the training set and development set first decrease and then remain flat, which can be seen as a reference for normal training process in the study. Figure 1c and 1d show the same metrics during the optimization process with $\mathcal{L}_{\text{a\&u}}$ and $\mathcal{L}_{\text{dcl}}$. Comparing with Figure 1b, the different variation trends on the training set and the development set imply some degree of overfitting. Specifically, when $\mathcal{L}_{\text{a\&u}}$ is applied for optimization, all three metrics except $\mathcal{L}_{\text{align}}$ show first decrease and then increase in the development set, while the similar situation occurs in all three metrics when $\mathcal{L}_{\text{dcl}}$ is applied.

### 3.2 OPTIMIZATION DYNAMICS STUDY

Since $\mathcal{L}_{\text{a\&u}}$ and $\mathcal{L}_{\text{dcl}}$ can be treated as equivalent to $\mathcal{L}_{\text{cl}}$ only when negative samples tend to infinity, we need to pay attention to what exactly is different between them when the number of negative samples is small. To gain more insights on their differences, we investigate the gradient property between contrastive loss and its decoupled forms in optimization dynamics.

The gradient of $\mathcal{L}_{\text{cl}}$ for the anchor $h_i$ can be split into two terms:

$$\frac{\partial \mathcal{L}_{\text{cl}}}{\partial h_i} = \underbrace{-\frac{1}{\tau} \frac{\sum_{j,j\neq i}^{N} \exp\left(\hat{h}_i^T \hat{h}_j'/\tau\right)\left(\hat{h}_i' - \hat{h}_i\right)}{\exp\left(\hat{h}_i^T \hat{h}_i'/\tau\right) + \sum_{j,j\neq i}^{N}\exp\left(\hat{h}_i^T \hat{h}_j'/\tau\right)} \frac{I - M_{h_i}}{\|h_i\|}}_{\nabla_{\text{cl}}^{\text{pos}}} \underbrace{-\frac{1}{\tau} \frac{\sum_{j,j\neq i}^{N} \exp\left(\hat{h}_i^T \hat{h}_j'/\tau\right)\left(\hat{h}_i - \hat{h}_j'\right)}{\exp\left(\hat{h}_i^T \hat{h}_i'/\tau\right) + \sum_{j,j\neq i}^{N}\exp\left(\hat{h}_i^T \hat{h}_j'/\tau\right)} \frac{I - M_{h_i}}{\|h_i\|}}_{\sum_{j,j\neq i}^{N} \nabla_{\text{cl}}^{\text{neg}\,j}} \quad (6)$$

where $I$ is the identity matrix and $M_{h_i}$ is the projection matrix on $h_i$.

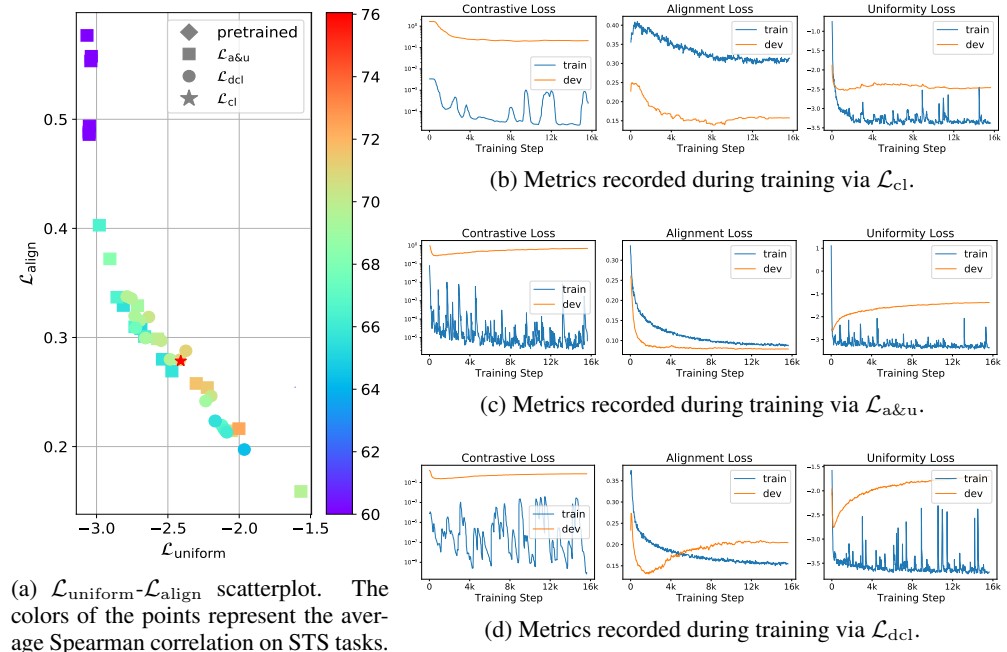

(a) $\mathcal{L}_{\mathrm{uniform}}$-$\mathcal{L}_{\mathrm{align}}$ scatterplot. The colors of the points represent the average Spearman correlation on STS tasks.

(b) Metrics recorded during training via $\mathcal{L}_{\mathrm{cl}}$.

(c) Metrics recorded during training via $\mathcal{L}_{\mathrm{a\&u}}$.

(d) Metrics recorded during training via $\mathcal{L}_{\mathrm{dcl}}$.

Figure 1: The difference between contrastive loss and its two decoupled forms in the training and evaluation phase. All above images are plotted based on the optimization process for $\mathrm{BERT_{base}}$.

When training is driven by such a gradient signal, the first term $\nabla_{\mathrm{cl}}^{\mathrm{pos}}$ points from $\hat{h}_i$ to $\hat{h}'_i$, pulling the anchor and the positive sample close to optimize *alignment*, while each $\nabla_{\mathrm{cl}}^{\mathrm{neg}_j}$ in the second term points from $\hat{h}'_j$ to $\hat{h}_i$, pushing the anchor away from the negative samples to optimize *uniformity*. Then the gradient norm on *alignment* is computed:

$$\left\| \nabla_{\mathrm{cl}}^{\mathrm{pos}} \right\| = \frac{1}{\tau} \frac{\sum_{j,j\neq i}^{N} \exp\left(\cos\theta_{ij'}/\tau\right) \left\| \hat{h}'_i - \cos\theta_{ii'}\hat{h}_i \right\|}{\exp\left(\cos\theta_{ii'}/\tau\right) + \sum_{j,j\neq i}^{N} \exp\left(\cos\theta_{ij'}/\tau\right)} \frac{1}{\|h_i\|} = \frac{1}{\tau} \frac{\sum_{j,j\neq i}^{N} \exp\left(\cos\theta_{ij'}/\tau\right) \sin\theta_{ii'}}{\exp\left(\cos\theta_{ii'}/\tau\right) + \sum_{j,j\neq i}^{N} \exp\left(\cos\theta_{ij'}/\tau\right)} \frac{1}{\|h_i\|} \quad (7)$$

where $\theta_{ii'}$ represents the angle between the anchor $\hat{h}_i$ and the positive sample $\hat{h}'_i$, and $\theta_{ij'}$ represents the angle between the anchor $\hat{h}_i$ and the negative sample $\hat{h}'_j$. Likewise, we can calculate the gradient norm of $\mathcal{L}_{\mathrm{align}}$ and $\mathcal{L}_{\mathrm{dcl}}$ on *alignment*, which are expressed by $\|\nabla\mathcal{L}_{\mathrm{align}}\|$ and $\|\nabla_{\mathrm{dcl}}^{\mathrm{pos}}\|$ separately:

$$\left\| \frac{\partial \mathcal{L}_{\mathrm{align}}}{\partial h_i} \right\| = \|\nabla\mathcal{L}_{\mathrm{align}}\| = \frac{\alpha(2\sin(\theta_{ii'}/2))^{\alpha-2}}{\|h_i\|} \left\| \hat{h}'_i - \cos\theta_{ii'}\hat{h}_i \right\| = \frac{\alpha(2\sin(\theta_{ii'}/2))^{\alpha-2}sin\theta_{ii'}}{\|h_i\|} \quad (8)$$

$$\|\nabla_{\mathrm{dcl}}^{\mathrm{pos}}\| = \frac{1}{\tau\|h_i\|} \left\| \hat{h}'_i - \cos\theta_{ii'}\hat{h}_i \right\| = \frac{\sin\theta_{ii'}}{\tau\|h_i\|} \quad (9)$$

Comparing equation 7, 8 and 9, we find that the gradient directions of these three parts are the same, but a key difference among them is that $\theta_{ij'}$ is contained in $\|\nabla_{\mathrm{cl}}^{\mathrm{pos}}\|$ but not in $\|\nabla\mathcal{L}_{\mathrm{align}}\|$ neither in $\|\nabla_{\mathrm{dcl}}^{\mathrm{pos}}\|$, which leads to a difference in whether the gradient signal is related to $\theta_{ij'}$.

Figure 2a visualizes this difference by plotting the heatmaps of $\|\nabla_{\mathrm{cl}}^{\mathrm{pos}}\|$, $\|\nabla\mathcal{L}_{\mathrm{align}}\|$ and $\|\nabla_{\mathrm{dcl}}^{\mathrm{pos}}\|$ with $\theta_{ii'}$-$\theta_{ij'}$. The leftmost four plots show the $\|\nabla_{\mathrm{cl}}^{\mathrm{pos}}\|$ under the different number of negative samples, while the rightmost two plots show $\|\nabla\mathcal{L}_{\mathrm{align}}\|$ and $\|\nabla_{\mathrm{dcl}}^{\mathrm{pos}}\|$ under the condition of 64 negative sample size. Observing the leftmost four plots, we can find that the area of the shaded part (or the area with weak gradient signals) gradually decreases as the number of negative sample increases. Therefore, it is conceivable that when the number of negative samples tends to infinity, these images will gradually become identical to the two rightmost images.

Comparing the leftmost plot with the rightmost two plots, we can clearly see the difference between them, where the former has a "gradient dissipation" situation that the latter does not have. So let us think what will happen during training. $\theta_{ij'}$ will increase and $\theta_{ii'}$ will decrease gradually. For

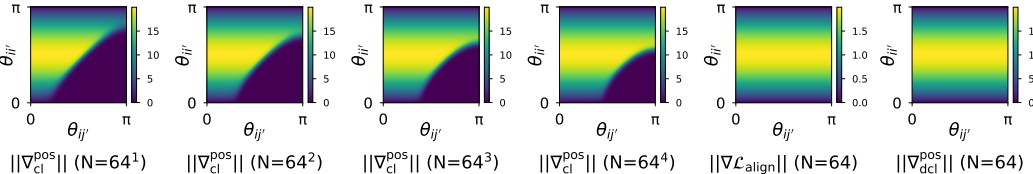

(a) Comparison among contrastive loss and its decoupled forms in the *alignment* part of the gradient norm.

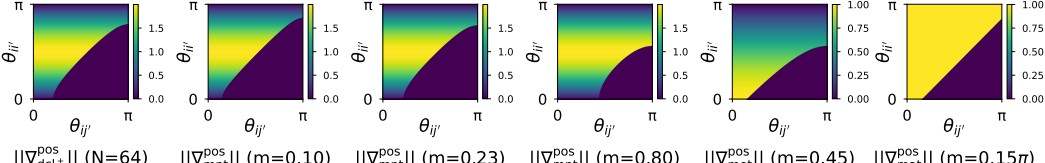

(b) Comparison among the proposed loss functions for validation in the *alignment* part of the gradient norm.

Figure 2: Gradient norms of contrastive loss and its two decoupled forms with respect to the *alignment* part. The $1/\|h_i\|$ in all equations are ignored when the images are plotted. All $\theta_{ij'}$ in one equation are treated as the same value. All hyper-parameters in equations are consistent with the columns corresponding to BERT$_{\text{base}}$ in Table 2.

$\|\nabla_{\text{cl}}^{\text{pos}}\|$, the gradient signals of some anchors will suddenly dissipate at a certain training phase and only the anchors whose $\theta_{ij'}$ is not too larger than $\theta_{ii'}$ will still receive the gradient signals. For $\|\nabla\mathcal{L}_{\text{align}}\|$ and $\|\nabla_{\text{dcl}}^{\text{pos}}\|$, since they are not constrained by $\theta_{ij'}$, their gradient signals will not vary with the horizontal axis. All anchors except those with absolutely small $\theta_{ii'}$ will receive the continuous gradient signals during training.

We point out that the above qualitative conclusion does not fail to hold if the angle is replaced with a generic distance function $\rho(.,.)$ on the hypersphere. Then the difference on "gradient dissipation" can be intuitively described as: "When the number of negative samples is small, contrastive loss tries to keep $\rho(\hat{h}_i, \hat{h}'_i)$ small relative to $\rho(\hat{h}_i, \hat{h}'_j)$ by some value, while its decoupled forms try to keep $\rho(\hat{h}_i, \hat{h}'_i)$ absolutely small". Here, we unthinkingly hypothesize that the "gradient dissipation" is a key property in the performance gap between contrastive loss and its decoupled forms, and more insights on this hypothesis will be provided in Section 5.

Due to the space limitation of the main text, only the gradients related to the *alignment* part are analyzed here. Please refer to Appendix B for the analysis of the *uniformity* part and Appendix C for the derivation of all equations in this section and the subsequent ones.

## 4 EXPERIMENTAL VERIFICATION

### 4.1 VALIDATION VIA THE INTRODUCTION OF $\mathcal{L}_{\text{dcl+}}$

To validate our hypothesis, we try to simulate this property with the smallest changes. Then two facts are noticed: (1) $\mathcal{L}_{\text{dcl}}$ will gradually become negative during training; (2) the "gradient dissipation" only occurs when $\theta_{ij'}$ is larger enough with respect to $\theta_{ii'}$. These two facts inspire us to introduce a ReLU function (Glorot et al., 2011) on top of $\mathcal{L}_{\text{dcl}}$ to force the truncation of the gradient signal provided to the anchor when $\mathcal{L}_{\text{dcl}} < 0$:

$$\mathcal{L}_{\text{dcl+}} = max\left(-\hat{h}_i^T\hat{h}'_i/\tau + \log\left(\sum_{j,j\neq i}^N \exp\left(\hat{h}_i^T\hat{h}'_j\right)/\tau\right), 0\right) \tag{10}$$

Likewise, we can obtain its gradient norm associated with *alignment*:

$$\left\|\nabla_{\text{dcl+}}^{\text{pos}}\right\| = \begin{cases} \dfrac{\sin\theta_{ij'}}{\tau}\dfrac{1}{\|h_i\|}, & -\hat{h}_i^T\hat{h}'_i/\tau + \log\sum_{j,j\neq i}^N \exp\left(\hat{h}_i^T\hat{h}'_j\right)/\tau > 0 \\ 0, & -\hat{h}_i^T\hat{h}'_i/\tau + \log\sum_{j,j\neq i}^N \exp\left(\hat{h}_i^T\hat{h}'_j\right)/\tau \leq 0 \end{cases} \tag{11}$$

The leftmost plot of Figure 2b shows $\left\|\nabla_{\text{dcl+}}^{\text{pos}}\right\|$ under the condition of 64 negative sample size. Comparing with the leftmost plot of Figure 2a, we can find that the images of $\left\|\nabla_{\text{dcl+}}^{\text{pos}}\right\|$ and $\left\|\nabla_{\text{cl}}^{\text{pos}}\right\|$

are almost identical, which proves that the added ReLU function can indeed simulate the property of "gradient dissipation". More importantly, the experimental results in Table 1 demonstrate the effectiveness of $\mathcal{L}_{\text{dcl+}}$, which obtains a 3%-7% improvement over $\mathcal{L}_{\text{dcl}}$ on STS tasks in the different pretrained models.

## 4.2 VALIDATION VIA THE UPPER BOUNDS FOR $\mathcal{L}_{\text{cl}}$

In fact, $\mathcal{L}_{\text{dcl+}}$ is introduced not only for the above mentioned observations, but also because $\mathcal{L}_{\text{dcl+}} + \log 2$ can be proved to be the upper bound of $\mathcal{L}_{\text{cl}}$. Further, we find that another upper bound with the property of "gradient dissipation", which can be derived after a second relaxation based on $\mathcal{L}_{\text{dcl+}}$:

$$
\begin{aligned}
\mathcal{L}_{\text{cl}} &\leq \log 2 + \max\left(-\hat{h}_i^T \hat{h}_i'/\tau + \log \sum_{j,j\neq i}^N \exp\left(\hat{h}_i^T \hat{h}_j'/\tau\right), 0\right) = \log 2 + \mathcal{L}_{\text{dcl+}} \\
&\leq \log 2 + \frac{1}{\tau} \max\left(-\hat{h}_i^T \hat{h}_i' + \max_{j,j\neq i}\left(\hat{h}_i^T \hat{h}_j\right) + \tau \log(N-1), 0\right)
\end{aligned}
\tag{12}
$$

where $\tau \log(N-1)$ can be replace as the margin hyperparameter $m$. Then we can get a new loss function:

$$
\mathcal{L}_{\text{mpt}} = \max\left(-\hat{h}_i^T \hat{h}_i' + \max_{j,j\neq i}\hat{h}_i^T \hat{h}_j + m, 0\right)
\tag{13}
$$

where the hardest negative sample is selected for optimization. Since this new loss function is very close in form to Triplet Loss (Weinberger & Saul, 2009), we express this loss function as $\mathcal{L}_{\text{mpt}}$ (Minimum dot Product Triplet Loss). To further increase the diversity of gradient norm variations, we replace the dot product with the Euclidean distance to obtain $\mathcal{L}_{\text{met}}$ (Minimum Euclidean distance Triplet Loss) and the angle to obtain $\mathcal{L}_{\text{mat}}$ (Minimum Angle Triplet Loss):

$$
\mathcal{L}_{\text{met}} = \max\{\|\hat{h}_i - \hat{h}_i'\|_2 - \min_{j,j\neq i}\|\hat{h}_i - \hat{h}_j'\|_2 + m, 0\}
\tag{14}
$$

$$
\mathcal{L}_{\text{mat}} = \max\{\theta_{ii'} - \min_{j,j\neq i}\theta_{ij'} + m, 0\}
\tag{15}
$$

Their performance on STS tasks and TR tasks are reported in Table 1 and find that the performance of these loss functions ahead of the decoupled forms of contrastive loss, and is comparable to or better than those of contrastive loss. Likewise, the gradient norm variations associated with *alignment*, $\left\|\nabla_{mpt}^{pos}\right\|$, $\|\nabla_{\text{met}}^{\text{pos}}\|$ and $\|\nabla_{\text{mat}}^{\text{pos}}\|$, are plotted in Figure 4c. As we can see, there are some difference among the gradient norms variations due to the choice of different distance functions, but it does not bring a significant difference to their performance, which proves that it is the "gradient dissipation" property and not a specific distance function that is at work.

## 4.3 MECHANISTIC ANALYSIS OF GRADIENT DISSIPATION

To gain a better understanding of the property of "gradient dissipation", we develop a interest in $\mathcal{L}_{\text{mpt}}$ because the degree of "gradient dissipation" can be adjusted by the only parameter $m$ (Figure 2b) and independent of the number of negative samples. With this more flexible form, how the margin value effects the performance can be quantitatively observed. Recalling that the "gradient dissipation" occurs when $d_{ij}$, i.e. $|\rho(\hat{h}_i, \hat{h}_i') - \min_{j,j\neq i} \rho(\hat{h}_i, \hat{h}_j')|$ goes outside a certain range and we can obtain different final $d_{ij}$ by selecting different $m$ during training. Figure 3a shows the performance on STS tasks and TR tasks with different $d_{ij}$. It can be clearly observed that the performance of STS tasks are much more sensitive to $d_{ij}$ than that of TR task. Specifically, too large or too small $d_{ij}$ will cause the performance of the STS task to drop sharply. In contrast, the performance on TR tasks does not vary significantly with $d_{ij}$, which seems to explain to some extent why the decoupled forms can work well in VRL. Surprisingly, no trade-offs are needed to guarantee the good performance on STS and TR tasks and a suitable margin value helps to achieve a double-best performance for both STS and TR tasks, which indicates that sentence representation quality can achieve both intrinsic and extrinsic excellence.

For a more in-depth look, we set $m$ to 0.10, 0.23 (corresponding the best performance) and 0.80 respectively, and record the numerical variation of $\mathcal{L}_{\text{cl}}$, $\mathcal{L}_{\text{align}}$ and $\mathcal{L}_{\text{uniform}}$ on the training and development sets. Comparing Figure 3c and 3d first, we find that $\mathcal{L}_{\text{cl}}$ can drop to lower on both training and validation sets when $m$ is 0.23, and this difference is magnified by the $\mathcal{L}_{\text{uniform}}$, while $\mathcal{L}_{\text{align}}$ remains almost constant. This observation is consistent with our intuition: when $m$ is too small, the gradient signals may be weak at the beginning of the training, leading to no improvement

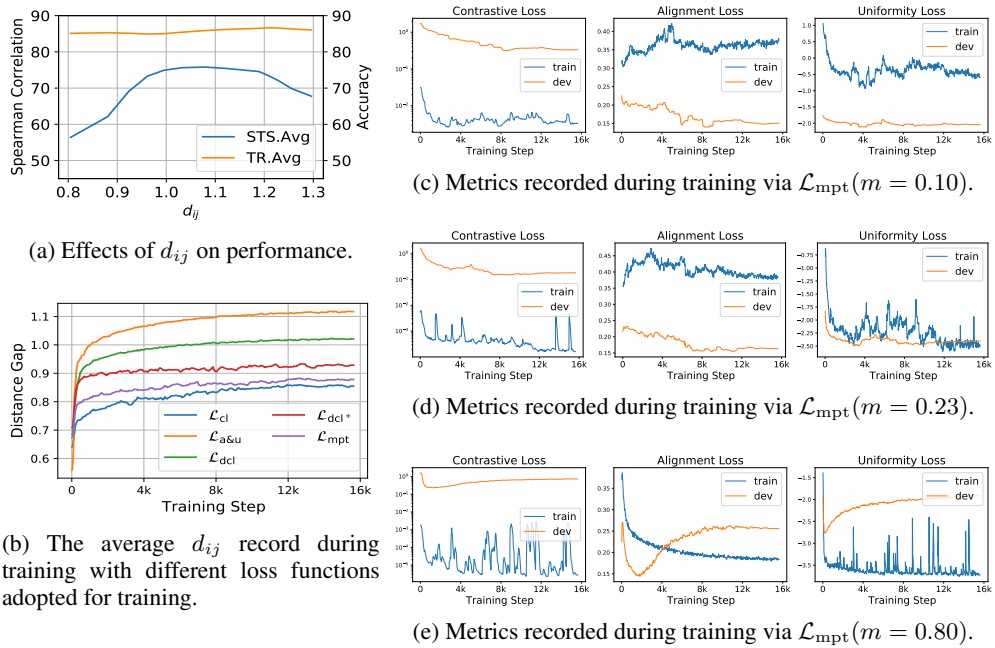

(a) Effects of $d_{ij}$ on performance.

(b) The average $d_{ij}$ record during training with different loss functions adopted for training.

(c) Metrics recorded during training via $\mathcal{L}_{\mathrm{mpt}}(m = 0.10)$.

(d) Metrics recorded during training via $\mathcal{L}_{\mathrm{mpt}}(m = 0.23)$.

(e) Metrics recorded during training via $\mathcal{L}_{\mathrm{mpt}}(m = 0.80)$.

Figure 3: The plots of the loss functions proposed in this work. $d_{ij}$ in (a) and (b) presents Euclidean distance. We first calculate the average distance of all negative samples to the anchor, then subtract the distance of positive samples to anchor from it to get the distance gap, and finally plot the average distance gap of all anchor in the mini-batch in the figure. All above images are plotted based on the optimization process for BERT$_{\mathrm{base}}$.

in the *uniformity* of the representation vectors. Then turn the attention to Figure 3e, which plots the same metrics when $m$ is 0.80, the curves as a whole is highly similar to Figure 1d, with the same overfitting phenomenon. We think it can be interpreted as an excessively large margin leads the optimization process exceptionally difficult, preventing the "gradient dissipation" phenomenon from occurring during training.

To summarize, the above phenomenons further illustrate that the timing of gradient dissipation is very important; too early "gradient dissipation" will result in under-tuned models, while too late "gradient dissipation" will result in degraded model performance and overfitting during training. We further study the connection between $\mathcal{L}_{\mathrm{mpt}}$ and the Mixup-based approaches (Kalantidis et al., 2020; Zhang et al., 2022b) in Appendix E.

## 5 DISCUSSION

In this section, we share some thoughts and observations to explain why we believe that the "gradient dissipation" plays a key role in SRL. We start by considering it in the relation to the STS tasks. For consistency with the retrieval scenario in practice, the STS tasks care about the orders among the semantic similarity of the anchors rather than the specific values among them (Reimers & Gurevych, 2019). In the ideal case, we should let the samples closer to the anchor if they are more semantically similar to the anchor, and no need to care about their absolute distance to the anchor. Intuitively, the property of "gradient dissipation" only works to puts negative samples further away from the anchor compared to positive samples, which is consistent with the need for STS tasks. On the other hand, *alignment* and *uniformity* only describe the relation between the anchor and its samples, while ignoring the relations between the positive samples and the negative samples. In other words, this optimization form overemphasizes the characteristics of the individual sentence itself and ignores the actual connections between the semantics. In early exploratory trials, we find that $d_{ij}$ is continuously enlarged when the decouple forms are adopted for training (Figure 3b), which is far beyond the level when trained by contrastive loss. Considering the limitations of data augmentation to generate

positive samples and the noise in sampling negative samples, we suspect that the large distance gap brought by the decouple forms compromises the original semantic information of the pretrained models to some extent.

Combining on the above thoughts and observations, we hypothesize that the "gradient dissipation" property is responsible for the better performance in the STS evaluation.

## 6 RELATED WORK

Sentence representation learning (SRL) (Kiros et al., 2015; Conneau et al., 2017; Reimers & Gurevych, 2019; Li et al., 2020) is one of the fundamental tasks in NLP, aiming at learning semantically rich high-dimensional representation at the sentence level. The good sentence representations need to satisfy both (1) high correlation with human-rated similarities (intrinsic evaluation) and (2) good transferability (extrinsic evaluation) (Chiu et al., 2016; Faruqui et al., 2016).

The pretrained language models (Devlin et al., 2018; Liu et al., 2019) were once regraded as the source for obtaining universal sentence representations, and the representations obtained by pretrained models have shown extraordinary performance on tranfer tasks. However, these representations obtained from the pretrained models even get lower performance than the average Glove embeddings (Pennington et al., 2014) on semantic textual similarity (STS) tasks (Reimers & Gurevych, 2019). Then more studies have found the word representation space of pretrained models, such as BERT (Devlin et al., 2018) and ELMo (Peters et al., 2018), are anisotropic (Ethayarajh, 2019), where word embeddings are concentrated on a high-dimensional conical space (Gao et al., 2018).

Early works (Li et al., 2020; Su et al., 2021) try to diminish the anisotropy of the pretrained representation space using the whitening transformation (Su et al., 2021) or the flow function (Li et al., 2020). These approaches are easy to implement, but limited improvement for STS tasks. Recently, contrastive learning methods based on instance discrimination (Zhang et al., 2022a;b;c) are introduced to SRL. However, a great deal of works (Yan et al., 2021; Gao et al., 2021; Zhou et al., 2022; Zhang et al., 2022b) focus on how to obtain or generate better positive and negative samples. At the same time, we find that even though there are significant differences in current contrastive learning methods in VRL and SRL, such as differences in pretraining and fine tuning, and differences in negative sample sizes (see Section 1 for details), few works have dabbled in the optimization mechanisms of contrastive learning in SRL. Instead, a large number of findings from VRL, such as *alignment* and *uniformity* (Wang & Isola, 2020; Gao et al., 2021), momentum updates (He et al., 2020; Wu et al., 2021) and bootstrap (Grill et al., 2020; Cao et al., 2022), have been directly applied.

Although there are a large number of works (Gao et al., 2021; Zhang et al., 2022b;c;a; Klein & Nabi, 2022) in SRL that use *alignment* and *uniformity* as evaluation metrics, to our knowledge, this work is the first work to study their inadequacy as the optimization objectives in SRL.

## 7 CONCLUSION

In this paper, we focus on the performance gap between contrastive loss and its decoupled forms, i.e., *alignment* and *uniformity* in SRL. Our series of new findings contribute to a deeper understanding of how contrastive loss can improve the quality of sentence representation on STS tasks: (1) *Alignment* and *Uniformity* Loss or their equivalent are not suitable as alternative loss functions for contrastive loss in SRL due to their lower performance and overfitting problem; (2) The "gradient dissipation" property of contrastive loss under a small number of negative samples plays a key role in the performance on STS tasks and preventing overfitting; (3) The "gradient dissipation" property works to control a suitable distance gap between anchor-positive and anchor-negative samples, while *alignment* and *uniformity* are the properties that control the absolute large distance of them; (4) Other two loss functions with "gradient dissipation" property also can solve the overfitting problem, and obtain the same or even better performance than contrastive loss, even if their "gradient dissipation" properties are not sensitive to the number of negative samples. We hope these findings to build a better understanding to the key properties of contrastive loss in SRL. Further, some new loss functions that are not bound to the form of contrastive loss can be designed via these properties, improving the quality of the sentence representation synthetically.

ACKNOWLEDGEMENT

This work was supported in part by the National Key R&D Program of China under Grant 2021ZD0110700, in part by the Fundamental Research Funds for the Central Universities, and in part by the State Key Laboratory of Software Development Environment.

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

## A  EXPERIMENT DETAILS

### A.1  EXPERIMENT SETUP

Following recent works in SRL (Gao et al., 2021; Yan et al., 2021; Zhang et al., 2022b), we use the SentEval toolkit (Conneau & Kiela, 2018) to evaluate the quality of sentence representations.

For intrinsic evaluation, i.e., correlation to human judgments, we use seven Semantic Textual Similarity (STS) datasets (STS12-16 (Agirre et al., 2012; 2013; 2014; 2015; 2016), STS-B (Cer et al., 2017) and SICK-R (Marelli et al., 2014) in SentEval. For extrinsic evaluation, i.e., transfer ability to downstream tasks, we also select seven downstream task datasets (MR (Pang & Lee, 2005), CR (Hu & Liu, 2004), SUBJ (Pang & Lee, 2004), MPQA (Wiebe et al., 2005), SST-2 (Socher et al., 2013), TREC (Voorhees & Tice, 2000) and MRPC (Dolan & Brockett, 2005)) to verify the migration ability of the sentence representations. We followed the default configuration recommended by SentEval for training and evaluation.

## A.2 TRAINING DETAILS

We validate the effectiveness of all loss functions based on the SimCSE (Gao et al., 2021) as it is effective but simple enough, so the performance is not easily influenced by other factors.

Following Gao et al. (2021), we observe the following points during training: (1) No data in any STS training set is used for training; (2) 1,000,000 sentences sampled from the English Wikipedia[2] are used for training instead; (3) Spearman correlation on STS-B development set is recorded each 125 steps; (4) The checkpoints corresponding to the highest Spearman correlation will be saved for evaluation; (5) The training period is one epoch for all pretrained models.

We implement the codes using Python3.7 and Pytorch1.12.0 and experiment with the single 32G NVIDIA V100 GPU.

| Method | Parameter | $BERT_{base}$ | $BERT_{large}$ | $RoBERTa_{base}$ | $RoBERTa_{large}$ |
|---|---|---|---|---|---|
| $\mathcal{L}_{cl}$ | $\tau$ | 0.05 | 0.05 | 0.05 | 0.05 |
| | learning rate | 1e-5 | 1e-5 | 1e-5 | 1e-5 |
| | batch size | 64 | 64 | 128 | 128 |
| $\mathcal{L}_{a\&u}$ | $\alpha$ | 2 | 2 | 2 | 2 |
| | t | 6 | 6 | 6 | 6 |
| | $\lambda$ | 0.1 | 0.1 | 0.2 | 0.4 |
| | learning rate | 1e-5 | 1e-5 | 1e-5 | 1e-5 |
| | batch size | 64 | 64 | 64 | 64 |
| $\mathcal{L}_{dcl}$ | $\tau$ | 0.03 | 0.01 | 0.02 | 0.02 |
| | learning rate | 1e-5 | 5e-5 | 1e-5 | 1e-5 |
| | batch size | 64 | 64 | 64 | 64 |
| $\mathcal{L}_{dcl+}$ | $\tau$ | 0.17 | 0.18 | 0.15 | 0.17 |
| | learning rate | 3e-5 | 5e-5 | 1e-5 | 1e-5 |
| | batch size | 64 | 64 | 64 | 64 |
| $\mathcal{L}_{mpt}$ | $m$ | 0.23 | 0.24 | 0.29 | 0.31 |
| | learning rate | 1e-5 | 1e-5 | 7e-6 | 7e-6 |
| | batch size | 64 | 64 | 64 | 64 |
| $\mathcal{L}_{met}$ | $m$ | 0.45 | 0.50 | 0.43 | 0.45 |
| | learning rate | 1e-5 | 1e-5 | 7e-6 | 7e-6 |
| | batch size | 128 | 128 | 128 | 128 |
| $\mathcal{L}_{mat}$ | $m$ | $0.15\pi$ | $0.17\pi$ | $0.14\pi$ | $0.13\pi$ |
| | learning rate | 1e-5 | 3e-5 | 7e-6 | 7e-6 |
| | batch size | 128 | 128 | 128 | 128 |

Table 2: The parameters corresponding to the best results of the STS tasks, which are also corresponding to the reported results and the plotted figures in this paper.

## A.3 PARAMETER SETTING

For all loss functions, we perform a grid search on learning rate =\{7e-6, 1e-5, 3e-5, 5e-5\} and batch size =\{64, 128, 256, 512\}. For other hyperparameters in every optimization objectives, we first narrow the interval with an extensive search, then the grid search is conducted in the following ranges:

- $\mathcal{L}_{cl}$ and $\mathcal{L}_{dcl}$
    - $\tau = \{0.03, 0.05, 0.07\}$
- $\mathcal{L}_{a\&u}$
    - $\lambda = \{0.1, 0.3, 0.5, 0.7, 0.9\}$

---

[2]https://huggingface.co/datasets/princeton-nlp/datasets-for-simcse/resolve/main/wiki1m_for_simcse.txt

- $\alpha = \{1.0, 2.0, 3.0\}$
- $t \in [2, 10]$, step size is 1

- $\mathcal{L}_{\text{dcl+}}$
  - $\tau \in [0.10, 0.20]$, step size is 0.01

- $\mathcal{L}_{\text{mpt}}$
  - $m \in [0.20, 0.35]$, step size is 0.01

- $\mathcal{L}_{\text{met}}$
  - $m \in [0.40, 0.50]$, step size is 0.01

- $\mathcal{L}_{\text{mat}}$
  - $m \in \{0.10\pi, 0.20\pi]$, step size is $0.01\pi$

The optimal parameters are shown in Table 2, which are adopted to report all results and plot to all plots in this paper.

## A.4 EVALUATION PROTOCOL

We report the performance of the sentence representation on the STS task using the Spearman's rank correlation, which has been widely used in recent works. Compared with the Pearson's correlation, Spearman's rank correlation focuses on relative ranking instead of absolute scores, which is more in line with practical retrieval applications of text similarity matching (Reimers et al., 2016).

## B THE ANALYSIS ON UNIFORMITY PART

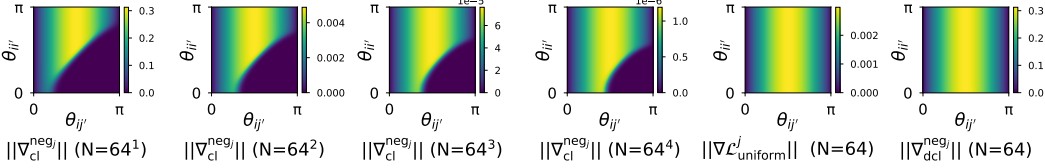

(a) Comparison among contrastive loss and its decoupled forms in the *uniformity* part of the gradient norm.

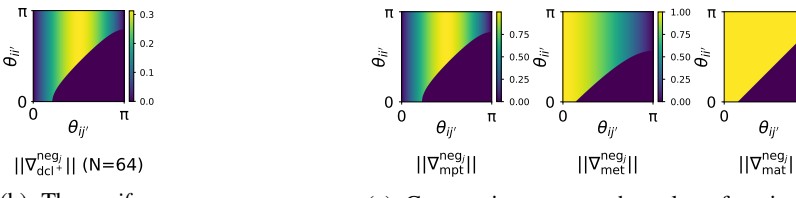

(b) The *uniformity* part of the gradient norm for $\mathcal{L}_{\text{dcl+}}$.

(c) Comparsion among three loss functions like Tripet Loss in the *uniformity* part of the gradient norm. The horizontal axis represents the minimum angle between the anchor and the negative samples.

Figure 4: Gradient norms of all the loss functions studied in the main text with respect to the *uniformity* part. All plotted images ignore $1/\|h_i\|$ and reflect only the gradient contribution of a single negative sample to the anchor.

Similar to the main text, we study the *uniformity* part of the gradient norm of all the loss functions mentioned in the main text, and their formulas are shown below:

$$\left\|\nabla_{\text{cl}}^{\text{neg}_j}\right\| = \frac{1}{\tau} \frac{\exp\left(\cos\theta_{ij'}/\tau\right)\sin\theta_{ij'}}{\exp\left(\cos\theta_{ii'}/\tau\right) + \sum_{j,j\neq i}^{N}\exp\left(\cos\theta_{ij'}/\tau\right)} \frac{1}{\|h_i\|} \tag{16}$$

$$\left\|\nabla\mathcal{L}_{\text{uniform}}^{j}\right\| = \frac{2t\exp\left(2t\cos(\theta_{ij'})\right)\sin\theta_{ij'}}{\sum_{i}^{N}\sum_{j,j\neq i}^{N}\exp\left(2t\cos(\theta_{ij'})\right)} \frac{1}{\|h_i\|} \tag{17}$$

$$\left\|\nabla_{\text{dcl}}^{\text{neg}_j}\right\| = \frac{1}{\tau} \frac{\exp\left(\cos\theta_{ij'}/\tau\right)\sin\theta_{ij'}}{\sum_{j,j\neq i}^{N}\exp\left(\cos\theta_{ij'}/\tau\right)} \frac{1}{\|h_i\|} \tag{18}$$

$$\left\|\nabla_{\text{dcl+}}^{\text{pos}}\right\| = \begin{cases} \dfrac{1}{\tau}\dfrac{\exp\left(\cos\theta_{ij'}/\tau\right)\sin\theta_{ij'}}{\sum_{j,j\neq i}^{N}\exp\left(\cos\theta_{ij'}/\tau\right)}\dfrac{1}{\|h_i\|}, & \hat{h}_i^T\hat{h}_i'/\tau - \log\displaystyle\sum_{j,j\neq i}^{N}\exp\left(\hat{h}_i^T\hat{h}_j'\right)/\tau < 0 \\[2em] 0, & \hat{h}_i^T\hat{h}_i'/\tau - \log\displaystyle\sum_{j,j\neq i}^{N}\exp\left(\hat{h}_i^T\hat{h}_j'\right)/\tau \geq 0 \end{cases} \tag{19}$$

$$\left\|\nabla_{\text{mpt}}^{\text{pos}}\right\| = \begin{cases} \dfrac{\sin\left(\theta_{ij'}\right)}{\|h_i\|}, & \hat{h}_i^T\hat{h}_i' + \max_{j,j\neq i}\hat{h}_i^T\hat{h}_j < m \\[1.5em] 0, & \hat{h}_i^T\hat{h}_i' + \max_{j,j\neq i}\hat{h}_i^T\hat{h}_j \geq m \end{cases} \tag{20}$$

$$\left\|\nabla_{\text{met}}^{\text{pos}}\right\| = \begin{cases} \dfrac{\cos\left(\theta_{ij'}/2\right)}{\|h_i\|}, & \|\hat{h}_i - \hat{h}_i'\|_2 - \min_{j,j\neq i}\|\hat{h}_i - \hat{h}_j'\|_2 < m \\[1.5em] 0, & \|\hat{h}_i - \hat{h}_i'\|_2 - \min_{j,j\neq i}\|\hat{h}_i - \hat{h}_j'\|_2 \geq m \end{cases} \tag{21}$$

$$\left\|\nabla_{\text{mat}}^{\text{pos}}\right\| = \begin{cases} \dfrac{1}{\|h_i\|}, & \theta_{ii'} - \min_{j,j\neq i}\theta_{ij'} < m \\[1.5em] 0, & \theta_{ii'} - \min_{j,j\neq i}\theta_{ij'} \geq m \end{cases} \tag{22}$$

The derivation of all the above equations can be found in Appendix C, and their trends with $\theta_{ii'}$-$\theta_{ij'}$ are plotted in Figure 4. The findings of the *uniformity* part is highly similar to those of the *alignment* part:

- $\|\nabla_{\text{cl}}^{\text{neg}_j}\|$ approaches $\|\nabla\mathcal{L}_{\text{uniform}}^{j}\|$ and $\|\nabla_{\text{dcl}}^{\text{neg}_j}\|$ as the negative sample size increases.
- When the number of negative samples is small, "gradient dissipation" exists for $\|\nabla_{\text{cl}}^{\text{neg}_j}\|$, while $\|\nabla\mathcal{L}_{\text{uniform}}^{j}\|$ and $\|\nabla_{\text{dcl}}^{\text{neg}_j}\|$ do not exist.
- $\|\nabla_{\text{dcl+}}^{\text{neg}_j}\|$ exists the similar "gradient dissipation" property of $\|\nabla_{\text{cl}}^{\text{neg}_j}\|$.
- $\|\nabla_{\text{mpt}}^{\text{pos}}\|$, $\|\nabla_{\text{met}}^{\text{pos}}\|$ and $\|\nabla_{\text{mat}}^{\text{pos}}\|$ also exist the propety of "gradient dissipation", but differs from $\|\nabla_{\text{cl}}^{\text{neg}_j}\|$ in the trend of the gradient norms.

## C  FORMULA DERIVATION

### C.1  THE DERIVATION OF EQUATION 6

$$\begin{aligned}
\frac{\partial\mathcal{L}_{\text{cl}}}{\partial h_i} &= -\frac{1}{\tau}\left(\hat{h}_i' + \frac{\exp\left(\hat{h}_i^T\hat{h}_i'/\tau\right)\hat{h}_i' + \sum_{j,j\neq i}^{N}\exp\left(\hat{h}_i^T\hat{h}_j'/\tau\right)\hat{h}_j'}{\exp\left(\hat{h}_i^T\hat{h}_i'/\tau\right) + \sum_{j,j\neq i}^{N}\exp\left(\hat{h}_i^T\hat{h}_j'/\tau\right)}\right)\frac{\partial\hat{h}_i}{\partial h_i} \\[1em]
&= -\frac{1}{\tau}\frac{\sum_{j,j\neq i}^{N}\exp\left(\hat{h}_i^T\hat{h}_j'/\tau\right)\left(\hat{h}_i' - \hat{h}_j'\right)}{\exp\left(\hat{h}_i^T\hat{h}_i'/\tau\right) + \sum_{j,j\neq i}^{N}\exp\left(\hat{h}_i^T\hat{h}_j'/\tau\right)}\frac{I - M_{h_i}}{\|h_i\|} \\[1em]
&= -\frac{1}{\tau}\sum_{j,j\neq i}^{N}\frac{\exp\left(\hat{h}_i^T\hat{h}_j'/\tau\right)\left(\hat{h}_i' - \hat{h}_i - \left(\hat{h}_j' - \hat{h}_i\right)\right)}{\exp\left(\hat{h}_i^T\hat{h}_i'/\tau\right) + \sum_{j,j\neq i}^{N}\exp\left(\hat{h}_i^T\hat{h}_j'/\tau\right)}\frac{I - M_{h_i}}{\|h_i\|} \\[1em]
&= \underbrace{-\frac{1}{\tau}\frac{\sum_{j,j\neq i}^{N}\exp\left(\hat{h}_i^T\hat{h}_j'/\tau\right)\left(\hat{h}_i' - \hat{h}_i\right)}{\exp\left(\hat{h}_i^T\hat{h}_i'/\tau\right) + \sum_{j,j\neq i}^{N}\exp\left(\hat{h}_i^T\hat{h}_j'/\tau\right)}\frac{I - M_{h_i}}{\|h_i\|}}_{\nabla_{\text{cl}}^{\text{pos}}} \\[1em]
&\quad \underbrace{-\frac{1}{\tau}\frac{\sum_{j,j\neq i}^{N}\exp\left(\hat{h}_i^T\hat{h}_j'/\tau\right)\left(\hat{h}_i - \hat{h}_j'\right)}{\exp\left(\hat{h}_i^T\hat{h}_i'/\tau\right) + \sum_{j,j\neq i}^{N}\exp\left(\hat{h}_i^T\hat{h}_j'/\tau\right)}\frac{I - M_{h_i}}{\|h_i\|}}_{\sum_{j,j\neq i}^{N}\nabla_{\text{cl}}^{\text{neg}_j}}
\end{aligned} \tag{23}$$

## C.2 THE DERIVATION OF EQUATION 7 AND EQUATION 16

$$\|\nabla_{\mathrm{cl}}^{\mathrm{pos}}\| = \left\| \frac{1}{\tau} \frac{\sum_{j,j\neq i}^{N} \exp\left(\hat{h}_i^T \hat{h}_j'/\tau\right)\left(\hat{h}_i' - \hat{h}_i\right)}{\exp\left(\hat{h}_i^T \hat{h}_i'/\tau\right) + \sum_{j,j\neq i}^{N} \exp\left(\hat{h}_i^T \hat{h}_j'/\tau\right)} \frac{I - M_{h_i}}{\|h_i\|} \right\|$$

$$= \frac{1}{\tau} \frac{\sum_{j,j\neq i}^{N} \exp\left(\hat{h}_i^T \hat{h}_j'/\tau\right)}{\exp\left(\hat{h}_i^T \hat{h}_i'/\tau\right) + \sum_{j,j\neq i}^{N} \exp\left(\hat{h}_i^T \hat{h}_j'/\tau\right)} \frac{\left\|\left(\hat{h}_i' - \hat{h}_i\right)\left(I - M_{h_i}\right)\right\|}{\|h_i\|}$$

$$= \frac{1}{\tau} \frac{\sum_{j,j\neq i}^{N} \exp\left(\cos\theta_{ij'}/\tau\right)}{\exp\left(\cos\theta_{ii'}/\tau\right) + \sum_{j,j\neq i}^{N} \exp\left(\cos\theta_{ij'}/\tau\right)} \frac{\left\|\hat{h}_i' - \hat{h}_i\cos\theta_{ij'}\right\|}{\|h_i\|} \tag{24}$$

$$= \frac{1}{\tau} \frac{\sum_{j,j\neq i}^{N} \exp\left(\cos\theta_{ij'}/\tau\right)\sin\theta_{ii'}}{\exp\left(\cos\theta_{ii'}/\tau\right) + \sum_{j,j\neq i}^{N} \exp\left(\cos\theta_{ij'}/\tau\right)} \frac{1}{\|h_i\|}$$

$$\left\|\nabla_{\mathrm{cl}}^{\mathrm{neg}_j}\right\| = \left\| \frac{1}{\tau} \frac{\exp\left(\hat{h}_i^T \hat{h}_j'/\tau\right)\left(\hat{h}_i - \hat{h}_j'\right)}{\exp\left(\hat{h}_i^T \hat{h}_i'/\tau\right) + \sum_{j,j\neq i}^{N} \exp\left(\hat{h}_i^T \hat{h}_j'/\tau\right)} \frac{I - M_{h_i}}{\|h_i\|} \right\|$$

$$= \frac{1}{\tau} \frac{\exp\left(\hat{h}_i^T \hat{h}_j'/\tau\right)}{\exp\left(\hat{h}_i^T \hat{h}_i'/\tau\right) + \sum_{j,j\neq i}^{N} \exp\left(\hat{h}_i^T \hat{h}_j'/\tau\right)} \frac{\left\|\left(\hat{h}_i - \hat{h}_j'\right)\left(I - M_{h_i}\right)\right\|}{\|h_i\|}$$

$$= \frac{1}{\tau} \frac{\exp\left(\cos\theta_{ij'}/\tau\right)}{\exp\left(\cos\theta_{ii'}/\tau\right) + \sum_{j,j\neq i}^{N} \exp\left(\cos\theta_{ij'}/\tau\right)} \frac{\left\|\hat{h}_i\cos\theta_{ij'} - \hat{h}_j'\right\|}{\|h_i\|} \tag{25}$$

$$= \frac{1}{\tau} \frac{\exp\left(\cos\theta_{ij'}/\tau\right)\sin\theta_{ij'}}{\exp\left(\cos\theta_{ii'}/\tau\right) + \sum_{j,j\neq i}^{N} \exp\left(\cos\theta_{ij'}/\tau\right)} \frac{1}{\|h_i\|}$$

## C.3 THE DERIVATION OF EQUATION 8 AND 17

$$\mathcal{L}_{\mathrm{align}} = \left(\left\|\hat{h}_i - \hat{h}_i'\right\|_2^2\right)^{\frac{\alpha}{2}} = \left(2 - 2\hat{h}_i^T \hat{h}_i'\right)^{\frac{\alpha}{2}} \tag{26}$$

$$\frac{\partial \mathcal{L}_{\mathrm{align}}}{\partial h_i} = \frac{\alpha}{2}\left(2 - 2\hat{h}_i^T \hat{h}_i'\right)^{\frac{\alpha-2}{2}}(-2\hat{h}_i')\frac{I - M_{h_i}}{\|h_i\|}$$

$$= -\alpha\left(2 - 2\hat{h}_i^T \hat{h}_i'\right)^{\frac{\alpha-2}{2}}\hat{h}_i'\frac{I - M_{h_i}}{\|h_i\|} \tag{27}$$

$$\left\|\frac{\partial \mathcal{L}_{\mathrm{align}}}{\partial h_i}\right\| = \alpha(2 - 2\cos\theta_{ii'})^{\frac{\alpha-2}{2}}\frac{\left\|\hat{h}_i' - \hat{h}_i\cos\theta_{ii'}\right\|}{\|h_i\|}$$

$$= \alpha(4\sin^2(\theta_{ii'}/2))^{\frac{\alpha-2}{2}}\sin\theta_{ii'}\frac{1}{\|h_i\|} \tag{28}$$

$$= \alpha(2\sin(\theta_{ii'}/2))^{\alpha-2}\sin\theta_{ii'}\frac{1}{\|h_i\|}$$

$$= \|\nabla\mathcal{L}_{\mathrm{align}}\|$$

$$\mathcal{L}_{\mathrm{uniform}} = \log\frac{1}{N(N-1)}\sum_{i}^{N}\sum_{j,j\neq i}^{N}\exp\left(-t\left\|\hat{h}_i - \hat{h}_j'\right\|_2^2\right) \tag{29}$$

$$\frac{\partial \mathcal{L}_{\mathrm{uniform}}}{\partial h_i} = \frac{-2t\sum_{j,j\neq i}^{N}\exp\left(-t\left\|\hat{h}_i - \hat{h}_j'\right\|_2^2\right)\left(\hat{h}_i - \hat{h}_j'\right)}{\sum_{i}^{N}\sum_{j,j\neq i}^{N}\exp\left(-t\left\|\hat{h}_i - \hat{h}_j'\right\|_2^2\right)}\frac{I - M_{h_i}}{\|h_i\|}$$

$$= \sum_{j,j\neq i}^{N}\nabla\mathcal{L}_{\mathrm{uniform}}^{j} \tag{30}$$

$$\left\| \nabla \mathcal{L}_{\text{uniform}}^{j} \right\| = \left\| \frac{2t \exp\left(-t \left\| \hat{h}_i - \hat{h}_j' \right\|_2^2\right) \left(\hat{h}_j' - \hat{h}_i\right)}{\sum_i^N \sum_{j,j\neq i}^N \exp\left(-t \left\| \hat{h}_i - \hat{h}_j' \right\|_2^2\right)} \frac{I - M_{h_i}}{\|h_i\|} \right\|$$

$$= \frac{2t \exp\left(-4t \sin^2(\theta_{ij'}/2)\right)}{\sum_i^N \sum_{j,j\neq i}^N \exp\left(-4t \sin^2(\theta_{ij'}/2)\right)} \frac{\left\| \left(\hat{h}_j' - \hat{h}_i\right)(I - M_{h_i}) \right\|}{\|h_i\|} \qquad (31)$$

$$= \frac{2t \exp\left(-4t \sin^2(\theta_{ij'}/2)\right)}{\sum_i^N \sum_{j,j\neq i}^N \exp\left(-4t \sin^2(\theta_{ij'}/2)\right)} \frac{\left\| \hat{h}_j' - \hat{h}_i \cos\theta_{ij'} \right\|}{\|h_i\|}$$

$$= \frac{2t \exp\left(-4t \sin^2(\theta_{ij'}/2)\right) \sin\theta_{ij'}}{\sum_i^N \sum_{j,j\neq i}^N \exp\left(-4t \sin^2(\theta_{ij'}/2)\right)} \frac{1}{\|h_i\|}$$

## C.4 The derivation of Equation 9, 11, 18 and 19

$$\frac{\partial \mathcal{L}_{\text{dcl}}}{\partial h_i} = - \underbrace{\frac{\hat{h}_i' - \hat{h}_i}{\tau} \frac{I - M_{h_i}}{\|h_i\|}}_{\nabla_{\text{dcl}}^{\text{pos}}}$$

$$- \underbrace{\frac{1}{\tau} \frac{\sum_{j,j\neq i}^N \exp\left(\hat{h}_i^T \hat{h}_j'/\tau\right)\left(\hat{h}_i - \hat{h}_j'\right)}{\sum_{j,j\neq i}^N \exp\left(\hat{h}_i^T \hat{h}_j'/\tau\right)} \frac{I - M_{h_i}}{\|h_i\|}}_{\sum_{j,j\neq i}^N \nabla_{\text{dcl}}^{\text{neg}\,j}} \qquad (32)$$

$$\left\| \nabla_{\text{dcl}}^{\text{pos}} \right\| = \left\| \frac{1}{\tau} \frac{\left(\hat{h}_j' - \hat{h}_i\right)(I - M_{h_i})}{\|h_i\|} \right\|$$

$$= \frac{1}{\tau} \frac{\left\| \left(\hat{h}_j' - \hat{h}_i\right)(I - M_{h_i}) \right\|}{\|h_i\|} \qquad (33)$$

$$= \frac{1}{\tau} \frac{\sin\theta_{ij'}}{\|h_i\|}$$

$$\left\| \nabla_{\text{dcl}}^{\text{neg}\,j} \right\| = \left\| \frac{1}{\tau} \frac{\exp\left(\hat{h}_i^T \hat{h}_j'/\tau\right)\left(\hat{h}_j' - \hat{h}_i\right)}{\sum_{j,j\neq i}^N \exp\left(\hat{h}_i^T \hat{h}_j'/\tau\right)} \frac{I - M_{h_i}}{\|h_i\|} \right\|$$

$$= \frac{1}{\tau} \frac{\exp\left(\hat{h}_i^T \hat{h}_j'/\tau\right)}{\sum_{j,j\neq i}^N \exp\left(\hat{h}_i^T \hat{h}_j'/\tau\right)} \frac{\left\| \left(\hat{h}_j' - \hat{h}_i\right)(I - M_{h_i}) \right\|}{\|h_i\|} \qquad (34)$$

$$= \frac{1}{\tau} \frac{\exp\left(\cos\theta_{ij'}/\tau\right)}{\sum_{j,j\neq i}^N \exp\left(\cos\theta_{ij'}/\tau\right)} \frac{\left\| \hat{h}_j' - \hat{h}_i \cos\theta_{ij'} \right\|}{\|h_i\|}$$

$$= \frac{1}{\tau} \frac{\exp\left(\cos\theta_{ij'}/\tau\right)\sin\theta_{ij'}}{\sum_{j,j\neq i}^N \exp\left(\cos\theta_{ij'}/\tau\right)} \frac{1}{\|h_i\|}$$

## C.5 THE DERIVATION OF EQUATION 12

$$
\begin{aligned}
\mathcal{L}_{\mathrm{cl}} &= -\log \frac{\exp\left(\hat{h}_i^T \hat{h}_i'/\tau\right)}{\exp\left(\hat{h}_i^T \hat{h}_i'/\tau\right) + \sum_{j,j\neq i}^N \exp\left(\hat{h}_i^T \hat{h}_j'/\tau\right)} \\
&= \log\left(1 + \frac{\sum_{j,j\neq i}^N \exp\left(\hat{h}_i^T \hat{h}_j'/\tau\right)}{\exp\left(\hat{h}_i^T \hat{h}_i'/\tau\right)}\right) \\
&= \log\left(1 + \exp\left(-\hat{h}_i^T \hat{h}_i'/\tau\right) \sum_{j,j\neq i}^N \exp\left(\hat{h}_i^T \hat{h}_j'/\tau\right)\right) \\
&= \log\left(1 + \exp\left(-\hat{h}_i^T \hat{h}_i'/\tau + \log \sum_{j,j\neq i}^N \exp\left(\hat{h}_i^T \hat{h}_j'/\tau\right)\right)\right) \\
&\leq \log 2 + \max\left(-\hat{h}_i^T \hat{h}_i'/\tau + \log \sum_{j,j\neq i}^N \exp\left(\hat{h}_i^T \hat{h}_j'/\tau\right), 0\right) \\
&\leq \log 2 + \max\left(-\hat{h}_i^T \hat{h}_i'/\tau + \log\left[(N-1)\max_{j,j\neq i}\exp\left(\hat{h}_i^T \hat{h}_j'/\tau\right)\right], 0\right) \\
&= \log 2 + \max\left(-\hat{h}_i^T \hat{h}_i'/\tau + \max_{j,j\neq i}\left(\hat{h}_i^T \hat{h}_j'\right)/\tau + \log(N-1), 0\right) \\
&= \log 2 + \frac{1}{\tau}\max\left(-\hat{h}_i^T \hat{h}_i' + \max_{j,j\neq i}\left(\hat{h}_i^T \hat{h}_j'\right) + \tau\log(N-1), 0\right)
\end{aligned}
\tag{35}
$$

where the first inequality sign holds due to the following inequality:

$$
\begin{aligned}
f(x) &= \log(1 + \exp(\mathrm{x})) \\
&= \log\left(\sum_{\mathrm{t}\in\{0,\mathrm{x}\}} \exp(\mathrm{t})\right) \\
&\leq \log(2\exp(\max(\mathrm{t}))) \\
&= \log 2 + \max(x, 0)
\end{aligned}
\tag{36}
$$

## C.6 THE DERIVATION OF $\left\|\nabla_{\mathrm{mpt}}^{\mathrm{pos}}\right\|$, $\left\|\nabla_{\mathrm{met}}^{\mathrm{pos}}\right\|$, $\left\|\nabla_{\mathrm{mat}}^{\mathrm{pos}}\right\|$, $\left\|\nabla_{\mathrm{mpt}}^{\mathrm{neg}}\right\|$, $\left\|\nabla_{\mathrm{met}}^{\mathrm{neg}}\right\|$ AND $\left\|\nabla_{\mathrm{mat}}^{\mathrm{neg}}\right\|$

Consider a generic form of the optimization objective first:

$$
\mathcal{L}_{\mathrm{mdt}} = \max\left(0, \rho\left(\hat{h}_i, \hat{h}_i'\right) - \min_{j,j\neq i}\rho\left(\hat{h}_i, \hat{h}_j'\right) + m\right)
\tag{37}
$$

where $\rho(.,.)$ represents a function used to calculate the similarity. Then the gradient of $\mathcal{L}_{\mathrm{mdt}}$ with respect to $h_i$ is calculated and decomposed:

$$
\frac{\partial \mathcal{L}_{\mathrm{mdt}}}{\partial h_i} = \begin{cases} \underbrace{\frac{\partial \rho\left(\hat{h}_i, \hat{h}_i'\right)}{\partial h_i}\frac{I - M_{h_i}}{\|h_i\|}}_{\nabla_{\mathrm{mdt}}^{\mathrm{pos}}} - \underbrace{\frac{\partial \min_{j,j\neq i}\rho\left(\hat{h}_i, \hat{h}_j'\right)}{\partial h_i}\frac{I - M_{h_i}}{\|h_i\|}}_{\nabla_{\mathrm{mdt}}^{\mathrm{neg}}}, & \rho\left(\hat{h}_i, \hat{h}_i'\right) - \min_{j,j\neq i}\rho\left(\hat{h}_i, \hat{h}_j'\right) < m \\ \\ 0, & \rho\left(\hat{h}_i, \hat{h}_i'\right) - \min_{j,j\neq i}\rho\left(\hat{h}_i, \hat{h}_j'\right) \geq m \end{cases}
\tag{38}
$$

where $\nabla_{\mathrm{mdt}}^{\mathrm{neg}}$ is only contributed by the negative sample with the closest distance to the anchor. By specifying $\rho(.,.)$ as dot product, $l_2$-norm and angle, we can get the gradient norm of $\mathcal{L}_{\mathrm{mpt}}$, $\mathcal{L}_{\mathrm{met}}$ and $\mathcal{L}_{\mathrm{mat}}$ with respect to the part of *alignment* respectively:

$$\|\nabla_{\text{mdt}}^{\text{pos}}\| = \begin{cases} \dfrac{\sin\left(\theta_{ii'}\right)}{\|h_i\|}, & \rho(\hat{h}_i, \hat{h}_i') = \hat{h}_i^T \hat{h}_i' \quad and \quad \rho\left(\hat{h}_i, \hat{h}_i'\right) - \min_{j,j\neq i}\rho\left(\hat{h}_i, \hat{h}_j'\right) < m \\[2mm] \dfrac{\cos\left(\theta_{ii'}/2\right)}{\|h_i\|}, & \rho(\hat{h}_i, \hat{h}_i') = -\left\|\hat{h}_i - \hat{h}_i'\right\|_2 \quad and \quad \rho\left(\hat{h}_i, \hat{h}_i'\right) - \min_{j,j\neq i}\rho\left(\hat{h}_i, \hat{h}_j'\right) < m \\[2mm] \dfrac{1}{\|h_i\|}, & \rho(\hat{h}_i, \hat{h}_i') = -\theta_{ij'} \quad and \quad \rho\left(\hat{h}_i, \hat{h}_i'\right) - \min_{j,j\neq i}\rho\left(\hat{h}_i, \hat{h}_j'\right) < m \\[2mm] 0, & \rho\left(\hat{h}_i, \hat{h}_i'\right) - \min_{j,j\neq i}\rho\left(\hat{h}_i, \hat{h}_j'\right) \geq m \end{cases} \tag{39}$$

Similarly, we can derive the gradient norm of each loss function corresponding to the part of *uniformity* respectively:

$$\|\nabla_{\text{mdt}}^{\text{neg}}\| = \begin{cases} \dfrac{\sin\left(\min\limits_{j,j\neq i}\theta_{ij'}\right)}{\|h_i\|}, & \rho(\hat{h}_i, \hat{h}_j') = \hat{h}_i^T \hat{h}_j' \quad and \quad \rho\left(\hat{h}_i, \hat{h}_i'\right) - \min_{j,j\neq i}\rho\left(\hat{h}_i, \hat{h}_j'\right) < m \\[3mm] \dfrac{\cos\left(\min\limits_{j,j\neq i}\theta_{ij'}/2\right)}{\|h_i\|}, & \rho(\hat{h}_i, \hat{h}_j') = -\left\|\hat{h}_i - \hat{h}_j'\right\|_2 \quad and \quad \rho\left(\hat{h}_i, \hat{h}_i'\right) - \min_{j,j\neq i}\rho\left(\hat{h}_i, \hat{h}_j'\right) \geq m \\[3mm] \dfrac{1}{\|h_i\|}, & \rho(\hat{h}_i, \hat{h}_j') = -\theta_{ij'} \quad and \quad \rho\left(\hat{h}_i, \hat{h}_i'\right) - \min_{j,j\neq i}\rho\left(\hat{h}_i, \hat{h}_j'\right) < m \\[3mm] 0, & \rho\left(\hat{h}_i, \hat{h}_i'\right) - \min_{j,j\neq i}\rho\left(\hat{h}_i, \hat{h}_j'\right) \geq m \end{cases} \tag{40}$$

## D   THE CONNECTION BETWEEN THE TWO DECOUPLED FORMS

In this paper, two important study objectives are two decoupled forms of contrastive loss. The first one is the *alignment* and *uniformity* proposed by Wang & Isola (2020):

$$\mathcal{L}_{\text{align}}\left(f; \alpha\right) \triangleq \mathop{\mathbb{E}}_{(\hat{h}_i, \hat{h}_i') \sim p_{\text{pos}}} \left[\|\hat{h}_i - \hat{h}_i'\|_2^\alpha\right], \quad \alpha > 0 \tag{41}$$

$$\mathcal{L}_{\text{uniform}}\left(f; t\right) \triangleq \log \mathop{\mathbb{E}}_{\substack{\text{i.i.d}\\(\hat{h}_i, \hat{h}_j') \sim p_{\text{data}}}} \left[\exp(-t\|\hat{h}_i - \hat{h}_j'\|_2^2)\right], \quad t > 0 \tag{42}$$

And the second one is decoupled contrastive loss (DCL, $\mathcal{L}_{\text{dcl}}$) proposed by Yeh et al. (2021):

$$\mathcal{L}_{\text{dcl}} = -\log \frac{\exp\left(\hat{h}_i^T \hat{h}_i'/\tau\right)}{\sum_{j,j\neq i}^N \exp\left(\hat{h}_i^T h_j'\right)/\tau} = -\hat{h}_i^T \hat{h}_i'/\tau + \log \sum_{j,j\neq i}^N \exp\left(\hat{h}_i^T h_j'\right)/\tau \tag{43}$$

where the first term of equation 43 is acknowledged by DCL's authors to be equivalent to equation 41, while the difference between the second term and $\mathcal{L}_{\text{uniform}}$ is only the order of the logarithmic function and the first summation operation when the losses are calculated for all samples in the same mini-batch. Their lower bounds can be obtained through Jensen's Inequality:

$$\mathcal{L}_{\text{uniform}} = \log \frac{1}{M(N-1)} \sum_i^M \sum_{j,j\neq i}^N \left[\exp(-t\|\hat{h}_i - \hat{h}_j'\|_2^2)\right]$$

$$\geq \frac{1}{M} \sum_i^M \log \frac{1}{N-1} \sum_{j,j\neq i}^N \left[\exp(-t\|\hat{h}_i - \hat{h}_j'\|_2^2)\right] \tag{44}$$

$$\geq -\frac{t}{M(N-1)} \sum_i^M \sum_{j,j\neq i}^N \|\hat{h}_i - \hat{h}_j'\|_2^2$$

For a mini-batch of data, the total loss is the mean value of the loss calculated for each anchor in the batch:

$$\mathcal{L}_{\text{dcl}}^{\text{neg}} = \log \sum_{j,j\neq i}^N \exp\left(\hat{h}_i^T h_j'\right)/\tau \tag{45}$$

$$
\begin{aligned}
\mathcal{L}_{\mathrm{dcl}}^{\mathrm{neg}} - \log(N-1) &= \frac{1}{M} \sum_{i}^{M} \log \frac{1}{N-1} \sum_{j, j \neq i}^{N} \exp(\hat{h}_i^T h_j')/\tau \\
&\geq \frac{1}{\tau M(N-1)} \sum_{i}^{M} \sum_{j, j \neq i}^{N} \hat{h}_i^T h_j' \\
&= -\frac{1}{2\tau M(N-1)} \sum_{i}^{M} \sum_{j, j \neq i}^{N} \|\hat{h}_i - \hat{h}_j'\|_2^2 + \frac{1}{2\tau M(N-1)}
\end{aligned}
\tag{46}
$$

where $M$ is batch size. Comparing equation 44 and 46, their common optimization objective is to maximize the sum of the squares of the Euclidean distance of all pairwise samples. Further, this optimization objective corresponds to a specific form of the minimum energy problem on the hypersphere (Kuijlaars & Saff, 1998; Liu et al., 2018), which is generalized from traditional Thomson Problem (Thomson, 1904) in physics.

## E  RELATION WITH MIXUP-BASED METHODS

We focus on a class of Mixup-based methods (Kalantidis et al., 2020; Zhang et al., 2022b) in contrastive learning which have similar optimization objectives to $\mathcal{L}_{\mathrm{mpt}}$. These methods use Mixup (Zhang et al., 2018) to generate hard negative samples for robustness and performance improvement, indicating the effeteness in VRL (Kalantidis et al., 2020) and SRL (Zhang et al., 2022b). It should to be noted that such methods are the improvements at the sample level, while do not change the property of "gradient dissipation" in contrastive loss.

Existing works generate mixup negative samples with two methods. The first one is generating by linearly weighting the representations of two hard negative samples (Kalantidis et al., 2020). If we regard $\hat{h}_i$ as the anchor, the mixup negative sample for the anchor can be defined as:

$$
\tilde{h}_k' = \lambda \hat{h}_m' + (1-\lambda)\hat{h}_n', \quad m \neq i, n \neq i
\tag{47}
$$

where $\hat{h}_m', \hat{h}_n'$ are two hard negative samples and $\tilde{h}_k'$ represents the mixup hard negative sample and $\lambda$ is the weight parameter. The second method is generating by linearly weighting the representations of a positive sample and a random negative sample (Kalantidis et al., 2020; Zhang et al., 2022b):

$$
\tilde{h}_k' = \lambda \hat{h}_i' + (1-\lambda)\hat{h}_k', \quad k \neq i
\tag{48}
$$

where $\hat{h}_k'$ is a random negative sample and $\lambda$ should be less than 0.5 to avoid generating pseudo-negative samples with high probability.

Then we discuss the relation between both two sample generation methods with $\mathcal{L}_{\mathrm{mpt}}$. When $M$ mixup negative samples are added to the original contrastive loss, the following inequality holds:

$$
\begin{aligned}
\mathcal{L}_{\mathrm{mix}} &= -\log \frac{\exp\left(\hat{h}_i^T \hat{h}_i'/\tau\right)}{\exp\left(\hat{h}_i^T \hat{h}_i'/\tau\right) + \sum_j^N \exp\left(\hat{h}_i^T \hat{h}_j'/\tau\right) + \sum_k^M \exp\left(\hat{h}_i^T \tilde{h}_k'/\tau\right)} \\
&= \log\left(1 + \frac{\sum_j^N \exp\left(\hat{h}_i^T \hat{h}_j'/\tau\right) + \sum_k^M \exp\left(\hat{h}_i^T \tilde{h}_k'/\tau\right)}{\exp\left(\hat{h}_i^T \hat{h}_i'/\tau\right)}\right) \\
&= \log\left(1 + \exp\left(-\hat{h}_i^T \hat{h}_i'/\tau\right)\left(\sum_j^N \exp\left(\hat{h}_i^T \hat{h}_j'/\tau\right) + \sum_k^M \exp\left(\hat{h}_i^T \tilde{h}_k'/\tau\right)\right)\right) \\
&= \log\left(1 + \exp\left(-\hat{h}_i^T \hat{h}_i'/\tau + \log\left(\sum_j^N \exp\left(\hat{h}_i^T \hat{h}_j'/\tau\right) + \sum_k^M \exp\left(\hat{h}_i^T \tilde{h}_k'/\tau\right)\right)\right)\right) \\
&\leq \log 2 + \max\left(-\hat{h}_i^T \hat{h}_i'/\tau + \log\left(\sum_j^N \exp\left(\hat{h}_i^T \hat{h}_j'/\tau\right) + \sum_k^M \exp\left(\hat{h}_i^T \tilde{h}_k'/\tau\right)\right), 0\right)
\end{aligned}
\tag{49}
$$

Then we need to perform a secondary relaxation, and the process needs to be discussed on the following sub-conditions:

**Condition 1** If $\tilde{h}'_k$ is generated using the way in equation 47, the maximum value of the $N + M$ terms in the log function must be taken in the first $N$ terms, and the following inequality holds:

$$
\begin{aligned}
\mathcal{L}_{\text{mix}} &\leq \log 2 + \max\left(-\hat{h}_i^T\hat{h}'_i/\tau + \log\left(\sum_j^N \exp\left(\hat{h}_i^T\hat{h}'_j/\tau\right) + \sum_k^M \exp\left(\hat{h}_i^T\tilde{h}'_k/\tau\right)\right), 0\right) \\
&\leq \log 2 + \max\left(-\hat{h}_i^T\hat{h}'_i/\tau + \max_j\left(\hat{h}_i^T\hat{h}_j\right)/\tau + \log(N+M), 0\right) \\
&= \log 2 + \frac{1}{\tau}\max\left(-\hat{h}_i^T\hat{h}'_i + \max_j\left(\hat{h}_i^T\hat{h}_j\right) + \tau\log(N+M), 0\right)
\end{aligned}
\tag{50}
$$

**Condition 2.1** If $\tilde{h}'_k$ is generated using the way in equation 48 and the maximum value of the $N+M$ terms in the log function is taken in the first $N$ terms, the inequality 50 still holds.

**Condition 2.2** If $\tilde{h}'_k$ is generated using the way in equation 48 and the maximum value of the $N+M$ terms in the log function is taken in the latter $M$ terms, the another inequality holds:

$$
\begin{aligned}
\mathcal{L}_{\text{mix}} &\leq \log 2 + \max\left(-\hat{h}_i^T\hat{h}'_i/\tau + \log\left(\sum_j^N \exp\left(\hat{h}_i^T\hat{h}'_j/\tau\right) + \sum_k^M \exp\left(\hat{h}_i^T\tilde{h}'_k/\tau\right)\right), 0\right) \\
&\leq \log 2 + \max\left(-\hat{h}_i^T\hat{h}'_i/\tau + \max_k\left(\hat{h}_i^T\tilde{h}_k\right)/\tau + \log(N+M), 0\right) \\
&= \log 2 + \max\left(-\hat{h}_i^T\hat{h}'_i/\tau + \max_k\left(\hat{h}_i^T\left(\lambda\hat{h}'_i + (1-\lambda)\hat{h}'_k\right)/\tau\right) + \log(N+M), 0\right) \\
&= \log 2 + \max\left(-\frac{1-\lambda}{\tau}\hat{h}_i^T\hat{h}'_i + \frac{1-\lambda}{\tau}\max_k\left(\hat{h}_i^T\hat{h}'_k\right) + \log(N+M), 0\right) \\
&= \log 2 + \frac{1-\lambda}{\tau}\max\left(-\hat{h}_i^T\hat{h}'_i + \max_k\left(\hat{h}_i^T\hat{h}'_k\right) + \frac{\tau}{1-\lambda}\log(N+M), 0\right)
\end{aligned}
\tag{51}
$$

These observations show the connection between $\mathcal{L}_{\text{mpt}}$ and the mixup-based contrastive methods: (1) if mixup negative samples are generated with the first method, the optimization objective of $\mathcal{L}_{\text{mpt}}$ will be equated to some extent with that of $\mathcal{L}_{\text{mix}}$ by appropriately adjusting $m$; (2) if mixup negative samples are generated with the second method, $\mathcal{L}_{\text{mix}}$ will have two upper bounds with different margins, $\tau\log(N+M)$ and $\frac{\tau}{1-\lambda}\log(N+M)$, during training. On the contrary, the margin $m$ in $\mathcal{L}_{\text{mpt}}$ will no longer change during training. Therefore, the optimization objective of $\mathcal{L}_{\text{mpt}}$ can be similar or equal to the mixup-based methods by using a appropriate $m$ (slightly larger than $\tau\log(N+M)$), which provides a new perspective to explain the well-performance of $\mathcal{L}_{\text{mpt}}$-like loss functions.

