# OpenReview forum: "On The Inadequacy of Optimizing Alignment and Uniformity in Contrastive Learning of Sentence Representations"
_ICLR.cc/2023/Conference — ICLR 2023 poster_

### Official Review · Reviewer_Rqm3 · 2022-10-25

**Confidence:** 2
**Clarity, Quality, Novelty And Reproducibility:** See above
**Correctness:** 3
**Technical Novelty And Significance:** 2
**Empirical Novelty And Significance:** 2
**Recommendation:** 5

**Strength And Weaknesses:**

Strengths:
- The paper investigates different variants of contrastive learning in the context of concrete tasks with natural language
- It is able to identify a problem with the decoupled contrastive learning approach in this setting, and propose a fix that both addresses this problem and restores performance on downstream tasks
- The paper proposed loss formulations close to the Triplet loss that typically achieve performance comparable or better than the InfoNCE loss

Weaknesses:
- After reading the paper, I didn't come away with a clear understanding regarding why the investigation into different contrastive losses is important. One way of looking at the situation is that for any given task there will always be some hyperparameter choices that perform better or worse than others, and in this situation the InfoNCE loss works well and the decoupled versions aren't as good. In that view, this key takeaway from the paper would simply be the advice to always use the InfoNCE objective for sentence representation learning. Clearly there is more going on, but I worry that my impression here points to potentially limited impact of the paper, or at least that the paper doesn't sufficiently convey the importance to the reader. For example, some of the past work was dealing with very real computational limitations related to large batch sizes, but that doesn't seem to be at issue here.
-

**Summary Of The Paper:**

The paper investigates contrastive learning in the context of building sentence representations. It finds that the decoupled form that decomposes the contrastive loss into aligment and uniformity components can have a detrimental effect in this setting, due to different training dynamics compared to the InfoNCE loss. These effects can be overcome by truncating the gradients once the negative examples are sufficiently further away compared to the positive example. Crucially, the amount of thresholding is controllable independently of the batch size.

**Summary Of The Review:**

Overall my assessment is that despite the very thorough experiments in the paper regarding different loss formulations, ultimately the scope of the paper is narrow compared to typical papers at ICLR.

---

> ### Author Response · Authors · 2022-11-17
> **Response to Reviewer Rqm3**
>
> Thank you for your comment, our replies to your concerns are as follows:
>
> **After reading the paper, I didn't come away with a clear understanding regarding why the investigation into different contrastive losses is important. One way of looking at the situation is that for any given task there will always be some hyperparameter choices that perform better or worse than others, and in this situation the InfoNCE loss works well and the decoupled versions aren't as good. In that view, this key takeaway from the paper would simply be the advice to always use the InfoNCE objective for sentence representation learning. Clearly there is more going on, but I worry that my impression here points to potentially limited impact of the paper, or at least that the paper doesn't sufficiently convey the importance to the reader. For example, some of the past work was dealing with very real computational limitations related to large batch sizes, but that doesn't seem to be at issue here.**
>
> We would like to point out that a fine grid search (Appendix A) is performed for the hyperparameters of all the loss functions involved in this paper, and the results reported in table 1 are the best results obtained for each loss function. For your other questions, we have answered them in the top comment in detail.

---

> > ### Author Response · Authors · 2022-12-09
> > **Discussion period ending**
> >
> > Dear reviewer, given that Discussion Stage 2 is approaching its end, we are wondering if our response has satisfactorily addressed your concerns. In case you have any further questions or comments, we will be glad to answer and discuss.

---

> ### Comment · Reviewer_Rqm3 · 2022-12-13
> **Re: framing**
>
> I realize that I have not left a comment in light of the discussion here.
>
> Based on the author responses, I think I have a better sense of what the purpose and motivation of the paper is. As I understand it, past attempts at applying contrastive learning for sentence representations in NLP have simply borrowed both methods and intuition from computer vision, without a deep investigation of whether the same same principles do in fact apply in both settings. One of the ways this has manifested itself is in terms of the prevalence of the alignment+uniformity decomposition, which has been justified in CV but the authors argue doesn't carry over to sentence representations. On a related point, contrastive learning in CV follows the principle of large batch sizes always being better, while in NLP it's typical to see smaller batch sizes though the paper doesn't go into the batch size discussion in too much detail.
>
> Ultimately, it is my view that the framing of the paper could be clearer, especially for readers without deep familiarity with the nuances of the contrastive learning literature across both CV and NLP. The question of the role of batch size (or effective batch size, to the extent that some methods attempt to imitate InfoNCE in a specific batch size regime), as well as whether this role is different for CV vs. NLP, appears to play a bigger role than the discussion devoted to it in the paper. I have decided to keep my overall score; I think a differently framed presentation of these results could be of value to the community, but I would not recommend acceptance of the paper as written.

---

> > ### Author Response · Authors · 2022-12-13
> > **Thank you**
> >
> > Thank you for your feedback.
> >
> > We agree that there may be room for improving the presentation of the paper, and we will certainly make an effort in that direction in the next revision. On the other hand, we feel that even in the current version, we have adequately delivered the message that contrastive learning in NLP (i.e. SRL) and in CV (i.e. VRL) reveals different behaviour and that small batch sizes in NLP make the alignment-uniformity decomposition an incorrect explanation for the effectiveness of infoNCE losses.  For example, in the abstract, we said explicitly " Considering the differences between VRL and SRL in terms of negative sample size and evaluation focus, we believe that the solid findings obtained in VRL may not be entirely carried over to SRL. In this work, we consider the suitability of the decoupled form of contrastive
> > loss, i.e., alignment and uniformity, in SRL. We find a performance gap between sentence representations obtained by jointly optimizing alignment and uniformity on the STS task and those obtained using contrastive loss". Also in Introduction, we said  "These observations suggest that alignment and uniformity losses might not serve suitable substitutes for contrastive loss in SRL and that the success of the standard contrastive learning for SRL can not be adequately explained in terms of alignment and uniformity properties or some delicate
> > balance between the two."  The paper also spends great effort explaining the impact of gradient dissipation in SRL. We think these messages are not ambiguous.
> >
> > If you have more specific suggestion as to how we may frame the paper differently for better presentation, we will be grateful to hear your input.

---

### Official Review · Reviewer_DG2P · 2022-10-29

**Confidence:** 3
**Correctness:** 4
**Technical Novelty And Significance:** 3
**Empirical Novelty And Significance:** 3
**Recommendation:** 6

**Clarity, Quality, Novelty And Reproducibility:**

*Clarity*
The paper was clearly written and I enjoyed reading the paper.

*Novelty*
Similar margin-based contrastive losses have been proposed [1,2,3]. I do not see how the contribution of this paper differs from previous papers which have used margin-based contrastive losses. If I am misunderstanding the contribution, I am happy to be corrected.

[1] Choi, Hongjun, Anirudh Som, and Pavan Turaga. "AMC-loss: Angular margin contrastive loss for improved explainability in image classification." Proceedings of the IEEE/CVF conference on computer vision and pattern recognition workshops. 2020.
[2] Hadsell, Raia, Sumit Chopra, and Yann LeCun. "Dimensionality reduction by learning an invariant mapping." 2006 IEEE Computer Society Conference on Computer Vision and Pattern Recognition (CVPR'06). Vol. 2. IEEE, 2006.
[3] Shah, Anshul, et al. "Max-Margin Contrastive Learning." Proceedings of the AAAI Conference on Artificial Intelligence. Vol. 36. No. 8. 2022.

**Strength And Weaknesses:**

*Weaknesses*
The contribution of a margin-based contrastive loss has been previously explored. See the Novelty section below.
Additionally, I find it concerning that BERT consistently outperforms RoBERTa in the downstream transfer tasks. Why is this the case?

*Additional questions and comments*
1. The bolding in Table 1 is not consistent across columns.
2. Is the reason gradient dissipation helps because it allows the model to focus on more difficult examples, rather than continuing to optimize easy ones which already satisfy the margin constraint?

**Summary Of The Paper:**

This paper identifies an issue with decomposed contrastive losses: a lack of gradient dissipation, which the paper claims leads to overfitting. Gradient dissipation means once negative examples are sufficiently far, those examples effectively no longer receive gradient. To fix this, a series of margin-based contrastive losses are proposed, with different energy functions: dot product, Euclidean distance, and angular difference. Models are then evaluated on semantic similarity tasks and other classification tasks (detailed in Appendix A.1).

**Summary Of The Review:**

I am not convinced that the proposed margin-based contrastive losses are novel, and found multiple papers that I found to be very similar [1,2,3].

---

> ### Author Response · Authors · 2022-11-17
> **Response to Reviewer DG2P Part Ⅱ**
>
> **3、The bolding in Table 1 is not consistent across columns.**
>
>
> We have done a more careful hyper-parameter search and obtained new results. Please see also "Response to All Reviewers".
>
>
> **4、Is the reason gradient dissipation helps because it allows the model to focus on more difficult examples, rather than continuing to optimize easy ones which already satisfy the margin constraint?**
>
> We think this view is in a sense correct. The gradient dissipation property effectively sets a margin between positive and negative samples to a certain degree, and disables the gradient signal for the negative examples after they are pushed beyond the margin. Then the gradient signal received at the negative examples may be seen as mainly acting on those "harder" negative examples, namely, those close to the positive sample (and within the margin). Indeed, as shown in Fig 3(b), "gradient dissipation" ensures that the negative examples maintain reasonable distance away from the positive examples but without having their gap being continuously maxmized.
>
> **Reference**
>
> [1] Choi, Hongjun, Anirudh Som, and Pavan Turaga. AMC-loss: Angular margin contrastive loss for improved explainability in image classification. CVPR2020.
>
> [2] Hadsell, Raia, Sumit Chopra, and Yann LeCun. Dimensionality reduction by learning an invariant mapping. CVPR2006.
>
> [3] Shah A, Sra S, Chellappa R, et al. Max-Margin Contrastive Learning. AAAI2022.
>
> [4] Caron M, Misra I, Mairal J, et al. Unsupervised learning of visual features by contrasting cluster assignments. NIPS2020.
>
> [5] Chen T, Kornblith S, Norouzi M, et al. A simple framework for contrastive learning of visual representations. ICML2020.
>
> [6] Tianyu Gao, Xingcheng Yao, and Danqi Chen. Simcse: Simple contrastive learning of sentence embeddings. EMNLP2021.
>
> [7] Liu Y, Ott M, Goyal N, et al. Roberta: A robustly optimized bert pretraining approach. arXiv preprint arXiv:1907.11692, 2019.

---

> > ### Comment · Reviewer_DG2P · 2022-11-20
> > **New loss functions => alternative loss functions**
> >
> > Thanks for the response. The contributions of 1) identifying a shortcoming of an approach directly taken from vision and applied to a new domain of text and 2) empirically and theoretically studying alternative loss functions in the new domain are valuable, and I will raise my score.
> >
> > However, I am not convinced that the loss functions are new. The functional form of a loss is the same whether it is used in supervised or unsupervised learning (where positive labels are generated via heuristics). Not proposing new losses does not lessen the contribution of analysis, and I recommend changing the prose to "analyzing alternative loss functions" rather than "proposing new loss functions".

---

> > > ### Author Response · Authors · 2022-12-12
> > > **Thank you very much !**
> > >
> > > We appreciate your positive feedback. Also, we will carefully consider your suggestions and fix the issues you mentioned in the new version of the paper.

---

> ### Author Response · Authors · 2022-11-17
> **Response to Reviewer DG2P Part Ⅰ**
>
> Thank you for your comment, our replies to your concerns are as follows:
>
> **1、Additionally, I find it concerning that BERT consistently outperforms RoBERTa in the downstream transfer tasks. Why is this the case?**
>
> This is indeed an interesting phenomenon. After comparing the evaluation protocol in the original RoBERTa paper[7] with ours, we find the conclusion of [7] suggesting that RoBERTa's tranfer ability is better than BERT is only based on the evaluation on GLUE Benchmark, SQuAD and RACE. And our TR datasets are selected from SentEval, where MR, CR, SUBJ, MPQA, and TREC are not included in GLUE. Interstingly, RoBERTa's results on MR, MPQA and TREC are 2%, 1% and 5% lower than BERT respectively, and its advantage on the remaining four datasets is rather small, causing the overall performance of RoBERTa to be lower than BERT.
>
> Notably the evaluation protocol in this work precisely follows that suggested by SentEval and is fully reproducible. Additionally experimental results giving similar observations are also reported in SimCSE, which can be found in Table E.1 of [6].
>
> **2、Novelty Similar margin-based contrastive losses have been proposed [1,2,3]. I do not see how the contribution of this paper differs from previous papers which have used margin-based contrastive losses. If I am misunderstanding the contribution, I am happy to be corrected.**
>
> As we state in "Response to All Reviewers", the main contribution of this paper is not the proposal of some novel loss functions. Although they do appear to bring some improvements (see the revised paper which includes updated results), the new loss functions primarily serve to support our analysis.
>
> We now highlight the difference between our paper and the papers [1, 2, 3] you brought up：
>
> As for [1,2], we note that the formula for the Contrastive loss (or Pair Loss) in their papers should be
>
> $\sum_{y_{ij}=1}\rho(x_i,x_j)+\sum_{y_{ij}=0}max(0, m-\rho(x_i,x_j))$
>
> where $\rho(.,.)$ is a distance function, $y_{ij}=1$ means $x_i$ and $x_j$ belong to the same category, and $y_{ij}=0$ means they belong to different categories. This loss function use in supervised metric learning widely.
>
> And our reference to ''contrastive loss'' in this paper actually refers to infoNCE Loss in unsupervised learning:
>
> $-\log \frac{\exp \left(\rho(x_i,x_j) / \tau\right)}{\left.\exp \left(\rho(x_i,x_j) / \tau\right)+\sum_{j,j \neq i}^{N} \exp \left(\rho(x_i,x_j)\right) / \tau\right)}$
>
> This designation is consistent with the recent literature [4,5]. What's more, the second class of loss functions we propose in this paper is actually an upper bound of InfoNCE Loss, which is actually closer to Triplet Loss in form., which also used in supervised metric learning widely.
>
> As for [3], the "max-margin" idea in [3] is exactly what we want to correct. "Max-margin" in contrastive learning means that the difference between the distance of negative samples from the anchor and the distance of positive samples from the anchor should be maximized. On the contrary, our findings suggest that the "margin" should not be maximized, but should be kept at an appropriate range, which is the key to the "gradient dissipation" property to work.
>
> The reasons for our conclusions that are inconsistent with those of [3] are as follows: As done in the majority of the current VRL papers, [3] only evaluated the transferability of representations on the downstream tasks (extrinsic evaluation). However, we evaluate the representation quality both on the STS tasks (intrinsic evaluation) and the TR tasks (extrinsic evaluation). As Table 1 shows, the results of $L_{\rm{a\\&u}}$ and $L_{\rm{dcl}}$ that seek to maximize margin are not significantly different from the results of the other rows on the TR tasks, while the difference is only significantly larger on the STS tasks.

---

### Official Review · Reviewer_gozg · 2022-10-29

**Confidence:** 3
**Correctness:** 2
**Technical Novelty And Significance:** 2
**Empirical Novelty And Significance:** 3
**Recommendation:** 6

**Clarity, Quality, Novelty And Reproducibility:**

Clarify: See strengths and weaknesses

Quality: See weaknesses

Reproducibility: All hyperparameter details are provided and the loss functions are easy to implement. It appears that the results can be reproduced with minimal effort.

**Strength And Weaknesses:**

Strengths
1. The paper shows interesting analysis and experiments showing the alternatives to InfoNCE proposed in the literature may not be as robust as previously presented, at least not outside of image representation learning.
2. The proposed new losses dcl+ and mpt are simple changes made to previous losses and still show improvements over baselines.
3. The experimental setup for different losses is thorough with extensive grid search done on different hyperparameters.

Weaknesses:
1. The gradient norm discussion in section 3.2 is not very convincing. In eq (6), the gradient wrt h_i is shown as a sum of a "positive" term \nabla^{pos} and a "negative" term \nabla^{neg} both of which have terms related to the anchor representation h_i. All the plots and justifications are given with respect the norms of these individual terms. It's not clear why it even makes sense to analyse the two terms separately given that the entire gradient term contains directions both of which update anchor-positive and anchor-negative relationships not just the infoNCE loss which has it in the "positive" term (in addition to the negative term"). I would prefer to see plots similar to figure (2) for the entire loss, not just individual components.

2. It is not clear entirely clear to me what the contribution of this work is. Is it a new loss function? is it showing that prior decoupled losses are not good for sentence representation learning? or something else? Additionally, the new losses presented in the paper still do not seem to outperform the infoNCE baseline for a larger (and much more widely used) model RoBERTa, so why should they be used?

3. Not too much of a concern but the writing is confusing in certain places (especially 3.2) and for example the paragraph right after eq (5) does not seem to serve a purpose.

**Summary Of The Paper:**

This paper analyzes previously proposed contrastive learning methods for image representation learning in the context of learning sentence representations. They consider the standard InfoNCE loss and two "decoupled" contrastive losses "A&U" (Wang and Isola 2020) and "DCL" (Yeh et al 2021) whose losses contain separate terms for alignment (decreasing the distance between an anchor and positive example) and uniformity (increasing the distance between the anchor and negative examples). They first show that the decoupled versions underperform InfoNCE and then present justification using gradients of the loss function with respect to the anchor representation. Their primary argument suggests that decoupled losses keep decreasing the alignment losses even when the negative examples are all far away whereas InfoNCE stops after a point. To counteract these issues, they propose two new losses akin to hinge losses and triplet losses which improve the performance to match the performance or outperform InfoNCE.

-------------------
Score updated after rebuttal.

**Summary Of The Review:**

While the paper shows the inadequacy of previously proposed alternatives to InfoNCE and proposes simple changes to address them however the optimization dynamics study is not entirely convincing of the justification of the introduced new losses. Further the new losses don't seem to improve over the baseline InfoNCE for larger more common underlying models such as RoBERTa. Hence I would not recommend acceptance in current form.

---

> ### Author Response · Authors · 2022-11-17
> **Response to Reviewer gozg**
>
> Thank you for your comment, our replies to your concerns are as follows:
>
> **1、The gradient norm discussion in section 3.2 is not very convincing. In eq (6), the gradient wrt $h_i$ is shown as a sum of a "positive" term $\nabla^{pos}$ and a "negative" term $\nabla^{neg}$ both of which have terms related to the anchor representation $h_i$. All the plots and justifications are given with respect the norms of these individual terms. It's not clear why it even makes sense to analyze the two terms separately given that the entire gradient term contains directions both of which update anchor-positive and anchor-negative relationships not just the infoNCE loss which has it in the "positive" term (in addition to the negative term"). I would prefer to see plots similar to figure (2) for the entire loss, not just individual components.**
>
> Without dividing the gradient into the two terms, the overall gradient of the contrastive loss for $h_i$ is a summation of the terms, where the gradient direction of each term points from some randomly sampled negative examples to the positive sample (Equation 23), making qualitative analysis difficult.
>
> On the other hand, after dividing the gradient into two terms, the physical meanings of both terms are clear: the "positive term" promotes alignment of the different views of the same example and the "negative term" promotes uniformity across all representations. Notably, similar decompositions of the gradient of the contrative loss haven applied and studies in the literature, see for example, [1,2].
>
> **2、It is not entirely clear to me what the contribution of this work is. Is it a new loss function? is it showing that prior decoupled losses are not good for sentence representation learning? or something else? Additionally, the new losses presented in the paper still do not seem to outperform the infoNCE baseline for a larger (and much more widely used) model RoBERTa, so why should they be used?**
>
> Please refer to "Response to All Reviewers".
>
> **3、Not too much of a concern but the writing is confusing in certain places (especially 3.2) and for example the paragraph right after eq (5) does not seem to serve a purpose.**
>
> We have rewritten the paragraph after Equation (5) for improved clarity. We emphasize that $L_{\rm{a\\&u}}$ and $L_{\rm{dcl}}$ are treated both decoupled forms of contrastive loss in this paper, since they both contain an alignment loss and a uniformity loss. In this sense, $L_{\rm{dcl}}$ is not an improved version of $L_{\rm{a\\&u}}$ as was claimed by the authors of $L_{\rm{dcl}}$.
>
> **Reference**
>
> [1] Wang T, Isola P. Understanding contrastive representation learning through alignment and uniformity on the hypersphere. ICML2020.
>
> [2] Zhang Y, Zhang R, Mensah S, et al. Unsupervised Sentence Representation via Contrastive Learning with Mixing Negatives. AAAI2022.

---

> > ### Author Response · Authors · 2022-12-09
> > **Discussion period ending**
> >
> > Dear reviewer, given that Discussion Stage 2 is approaching its end, we are wondering if our response has satisfactorily addressed your concerns. In case you have any further questions or comments, we will be glad to answer and discuss.

---

> > > ### Comment · Reviewer_gozg · 2022-12-11
> > > **Thanks for your response!**
> > >
> > > Thank you to the authors for their response. The concerns I raised in the review have mostly been resolved. So I am raising my score to 6.
> > >
> > > From the authors' comments, the overall contribution of the work is pointing out that constrastive learning for VRL and SRL has differences and the same assumptions that work for one may not work for the other. While this makes sense, it should be clearly stated in the abstract/introduction. Current version of the paper while alludes to this but does not reflect this clearly in terms of actual differences. I hope it will be clarified in a future version of the paper.

---

> > > > ### Author Response · Authors · 2022-12-12
> > > > **Thank you very much!**
> > > >
> > > > We appreciate your positive feedback. Also, we will carefully consider your suggestions and fix the issues you mentioned in the new version of the paper.

---

### Official Review · Reviewer_qhtV · 2022-11-01

**Confidence:** 3
**Correctness:** 3
**Technical Novelty And Significance:** 2
**Empirical Novelty And Significance:** 4
**Recommendation:** 6

**Clarity, Quality, Novelty And Reproducibility:**

The paper is generally well-written and provides several novel perspectives that can help us to better understand the inner workings of contrastive learning for SRL.

**Strength And Weaknesses:**

Strengths
- This work attempts to directly train SRL models based on alignment & uniformity which are usually used as diagnostic metrics to evaluate the effectiveness of SRL models, and from the trial, the authors reveal some intuitive findings on why contrastive learning works in SRL.
- The paper presents a series of experiments with the newly proposed learning objectives that persuasively back up the claim that the "gradient dissipation" property is a key to the success of contrastive learning for SRL.

Weaknesses
- Conditioned on Figure 2 (a), it seems that the "gradient dissipation" property appears especially in the case where the number of negative examples for contrastive learning is relatively small and that the property vanishes as the number of negative examples increases. Therefore, if the property is truly a key to success for contrastive learning, we would arrive at a weird conclusion that it is better to reduce the number of negative examples, which contradicts the wisdom in the literature that it is almost always better to adopt more negative examples.
- Although the storyline presented in the work was quite persuasive, it is still unclear whether we can make an assertive conclusion only relying on the information obtained from the gradient norm of models.
- From Table 1, while the proposed losses ($L_{mpt}, L_{met}$, and $L_{mat}$) are better than the typical contrastive learning ($L_{cl}$) when evaluated on BERT, they do not show any superiority when tested on RoBERTa, which weakens the authors' powers of persuasion.


**Summary Of The Paper:**

This paper aims to investigate the secrets of why contrastive learning for sentence representation learning (SRL) works from the perspective of alignment and uniformity (Wang & Isola, 2020).

While alignment & uniformity are usually utilized in the literature as a tool for demonstrating the effectiveness of some SRL approaches, this work attempts to directly optimize sentence embedding models based on the metrics, showing that this way of training results in inferior performance compared to that of typical contrastive learning.

In addition, the authors claim that one of the reasons why the performance gap exists is that contrastive learning exhibits the "gradient dissipation" property (especially when the number of negative examples is relatively small), which is not shown when one trains SRL models directly using alignment & uniformity.

To back up their claim, the authors propose several revised learning objectives that can simulate the "gradient dissipation" property even in the case we do not use the original form of contrastive learning and shows that such loss functions can improve the performance of SRL models.




**Summary Of The Review:**

To summarize, although the paper presents an attractive story that reveals a part of the secrets of why contrastive learning is effective in SRL, it is still a little bit unclear whether the claims made by the authors are absolutely technically correct, especially considering some concerns I explained in the above Weaknesses part.

---

> ### Author Response · Authors · 2022-11-17
> **Response to Reviewer qhtV**
>
> Thank you for your comment, our replies to your concerns are as follows:
>
> **1、Conditioned on Figure 2 (a), it seems that the "gradient dissipation" property appears especially in the case where the number of negative examples for contrastive learning is relatively small and that the property vanishes as the number of negative examples increases. Therefore, if the property is truly a key to success for contrastive learning, we would arrive at a weird conclusion that it is better to reduce the number of negative examples, which contradicts the wisdom in the literature that it is almost always better to adopt more negative examples.**
>
>
> In fact，it is consistent with the results in the recent studies. "More negative examples is better" is always correct in VRL, while the number of negative samples is usually set small in SRL. For example, the number of negative samples is used to being set 63\~160 for BERT[1,3,4,5] and 64\~511 for Roberta[1,4,5], while these number are even less in [2]. It is noteworthy to highlight that all these papers tried larger negative sample sizes and obtained worse results.
>
> Based on the above facts, we point out that $alignment$ and $uniformity$ are no longer applicable because they assume a large negative sample size, while the "gradient dissipation" property we found is more suitable to explain the working mechanism of contrastive learning with a small nagetive sample size.
>
> **2、Although the storyline presented in the work was quite persuasive, it is still unclear whether we can make an assertive conclusion only relying on the information obtained from the gradient norm of models.**
>
> It is arguably true that information revealed from the gradient norm may not reflect the full picture of contrastive learning for SRL. This is particularly because unsupervised representation learning is still at its infancy, and there are more unknowns in the problem scope than what has been understood. Nonetheless our experiments show a significant performance gap between loss functions with the "gradient dissipation" property and those without. This should suggest that the ''gradient dissipation'' property is at least a necessary condition for high performance when the number of negative examples is relative small.
>
> We start from the gradient norms because (1) the conclusion that the optimization objective of $\cal{L}_\rm{cl}$ can be decomposed into $alignment$ and $uniformity$ when the number of negative samples tends to infinity is clear; (2) the main difference between SRL and VRL in the training protocol is the smaller number of negative samples. So it is easy to think of plotting the image of gradient norms and studying the variation of the gradient signals with the distance between the archor and the sample.
>
> **3、From Table 1, while the proposed losses (Lmpt,Lmet, and Lmat) are better than the typical contrastive learning (Lcl) when evaluated on BERT, they do not show any superiority when tested on RoBERTa, which weakens the authors' powers of persuasion.**
>
> We have more carefully tuned the hyper-parameters for the proposed loss functions, and they also demonstrate superior performance on RoBERTa. Please refer to "Response to All Reviewers".
>
> **Reference**
>
> [1] Tianyu Gao, Xingcheng Yao, and Danqi Chen. Simcse: Simple contrastive learning of sentence embeddings. EMNLP2021.
>
> [2] Yuhao Zhang, Hongji Zhu, Yongliang Wang, Nan Xu, Xiaobo Li, and Binqiang Zhao. A contrastive framework for learning sentence representations from pairwise and triple-wise perspective in angular space. ACL2022.
>
> [3] Zhang Y, Zhang R, Mensah S, et al. Unsupervised Sentence Representation via Contrastive Learning with Mixing Negatives. AAAI2022.
>
> [4] Xing Wu, Chaochen Gao, Liangjun Zang, Jizhong Han, Zhongyuan Wang, and Songlin Hu. 2022. ESimCSE: Enhanced Sample Building Method for Contrastive Learning of Unsupervised Sentence Embedding. COLING2022.
>
> [5] Yan Y, Li R, Wang S, et al. ConSERT: A Contrastive Framework for Self-Supervised Sentence Representation Transfer. ACL2021.

---

> > ### Author Response · Authors · 2022-12-09
> > **Discussion period ending**
> >
> > Dear reviewer, given that Discussion Stage 2 is approaching its end, we are wondering if our response has satisfactorily addressed your concerns. In case you have any further questions or comments, we will be glad to answer and discuss.

---

### Author Response · Authors · 2022-11-17
**Response to All Reviewers**

We thank all reviewers sincerely for their valuable comments. Here we address some common issues raised by the reviewers.

**Research Motivation**

This research aims at understanding the role of InfoNCE loss in the unsupervised sentence representation learning (SRL). Since the work of [1], the working of InfoNCE loss in visual representation learning (VRL) has been understood via a decomposition into an alignment loss and a uniformity loss; and such an understanding is also taken for granted in the SRL literature to explain the effectiveness of InfoNCE loss for SRL.

We however note that SRL tasks differ significantly from the VRL tasks, in the context of contrastive learning. Specificallly, (1) The performance of SRL is insensitive to the number of negative samples. A small number of negative samples can achieve the same or better results with a large number of negative samples[2]; (2) The representation quality is evaluated on both intrinsic tasks (e.g., semantic textual similarity) and extrinsic tasks (e.g., classification)

The vast majority of current research in SRL ignores these differences and directly accepts the finding on $alignment$ and $uniformity$ from VRL, regrading them as evaluation metrics also for SRL. In this work, we challenge this understanding and explore the true reason why InfoNCE is effective in SRL.


**Paper Contributions**

The main contributions of this paper are as follows.
(1) We suggest that it is inadequate to use $alignment$ and $uniformity$ to explain the effectiveness of InfoNCE loss in SRL.
(2) We show that a ''gradient dissipation'' property plays an important role in InfoNCE Loss in unsupervised SRL.

**On the Performance of the Proposed Loss Functions**

We emphasize that the loss functions proposed in this paper were not to create SOTA models that outperform the InfoNCE Loss based models. Rather these loss functions primarily served to further demonstrate the effectiveness of the "gradient dissipation" property: The loss functions with the "gradient dissipation" property consistently outperform those without the property. For that reason, we did not carefully tune the hyper-parameter, thinking our experiments were adequate.

After seeing the reviewers's comments, we nonetheless made some effort searching for good hyper-parameters. As a consequence, our proposed loss functions have surpassed InfoNCE on both Base and Large versions of BERT and RoBERTa. The results we report on RoBERTa is the average of five runs, which include original three runs and new two runs, and their random seeds form an arithmetic sequence. We will replace all the results in Table 1 with the average of the five runs with the same random seeds as soon as possible.

**Paper Updates**

We have updated the version of the paper, corrected some language issues and gramma errors, and added some observations to help reviewers better understand the problem we study. The changes in the new version are highlighted in blue.

**Reference**

[1] Wang T, Isola P. Understanding contrastive representation learning through alignment and uniformity on the hypersphere. ICML2020.

[2] Tianyu Gao, Xingcheng Yao, and Danqi Chen. Simcse: Simple contrastive learning of sentence embeddings. EMNLP2021.

---

### Decision · Program_Chairs · 2023-01-20

**Decision:**

Accept: poster

**Justification For Why Not Higher Score:**

The limited scope of this work, as reviewers agreed during the meeting that it is narrow compared to typical work accepted at ICLR.

The framing of the paper is also somewhat confusing, and two of the reviewers were initially not sure what the paper was about. (During the meeting, the reviewers agreed there is a relative lack of focus and scope.)

**Justification For Why Not Lower Score:**

Despite the limited scope, this is a solid paper on representation learning.

**Metareview: Summary, Strengths And Weaknesses:**

The paper analyzes contrastive learning methods initially developed for visual representation learning (VRL) and that have since then been used for sentence representation learning (SRL). In particular, the authors analyze the decoupling of the contrastive loss into “alignment loss” and “uniformity loss” (theoretically justified by Wang and Isola, 2020 for VRL) and show that it is problematic in its application to SRL. Indeed, this decomposition holds when the number of negative examples approaches infinity, but contrastive learning for SRL is often more effective with small numbers of negative examples. To empirically demonstrate their claim, the authors show that the decoupled loss underperforms InfoNCE on SRL, and often quite significantly. The authors propose new losses based on the "gradient dissipation" property that outperform InfoNCE (substantially on BERT, but on a smaller margin on RoBERTa).

Strengths:
1. The paper demonstrates the inadequacy and ineffectiveness of alignment and uniformity losses in SRL. This finding is valuable considering that, e.g., prior work on SRL has attributed the success of contrastive learning on semantic text similarity tasks to a good balance between alignment and uniformity.
2. The gradient dissipation property seems to help. In its latest revision, the paper also provides positive results as it outperforms InfoNCE in all experimental conditions.

Weaknesses:
1. Unclear framing: As mentioned in two of the reviews, it is often unclear what the paper is about (analyses of alignment and uniformity losses? or proposal of new losses?). While the authors clarified their intentions in the response to all reviewers, the reviewers think the framing should be revisited, and contributions be made clearer in the abstract and introduction. In spite of these concerns expressed in the original reviews, the revised paper does not appear to improve the framing. The reviewers found this problematic as they felt the framing issue needs to be fixed for the paper to be considered for acceptance.
2. Narrow scope: Some reviewers considered the scope of the paper to be too narrow for ICLR, and this issue was discussed extensively at the reviewer-AC meeting. The paper particularly focuses on the decomposition principle between alignment and uniformity, and the reviewers felt the paper would have been much stronger if it had compared the application of contrastive learning across VRL and SRL in a more general manner. One reviewer also noted a problem of scope while discussing reviewer's qhtV concern that the number of negative samples is too small. Indeed, the paper questions one assumption (i.e., decomposition principle) while taking some other assumption for granted (e.g., that having few negative samples is preferable for SRL). While it may very well be true that having fewer negative examples helps in SRL, providing more analyzes of the effect of the number of negative samples (e.g., in the STS results) would have shed more light on the effectiveness of the methods evaluated in the paper, and helped reduce reviewer qhtV’s concerns about the small number of negative samples.

Overall, the reviewers found the paper to be quite borderline -- it provides some interesting analyzes and empirical gains, but the scope is relatively narrow. I think the strengths of the paper outweigh its weaknesses and I recommend acceptance.

**Note From Pc:**

if the above contains the word "oral" or "spotlight" please see: "oral" presentation means -> notable-top-5% and "spotlight" means -> notable-top-25%. As stated in our emails, we are disassociating presentation type from AC recommendations

**Summary Of Ac-Reviewer Meeting:**

Most of the meeting was spent on discussing "scope" and "framing" issues, which were significant concerns to the reviewers (but not the point of requiring rejection). They felt these cons equally weighed the two pros of the meta-review. They concluded this that this is a solid paper, but a bit of a niche paper. All reviewers agreed the paper is very borderline, and I didn't manage to get a consensus on either accept or reject. One leaned positively, one negatively, and the two others couldn't make their mind. I encouraged them to make a stand, but to no avail and they left it to the AC and SAC to make a recommendation.

Note: Reviewer qhtV (expert in contrastive learning) remained unconvinced by the author's response about the small number of negative examples, but the three other reviewers were satisfied by the response (except that this concern raised another one about scope.)